# Non-Local Latent Relation Distillation for Self-Adaptive 3D Human Pose Estimation

**Jogendra Nath Kundu**[1]    **Siddharth Seth**[1]    **Anirudh Jamkhandi**[1]    **Pradyumna YM**[1]
**Varun Jampani**[2]    **Anirban Chakraborty**[1]    **R. Venkatesh Babu**[1]

[1]Indian Institute of Science, Bangalore    [2]Google Research

## Abstract

Available 3D human pose estimation approaches leverage different forms of strong (2D/3D pose) or weak (multi-view or depth) paired supervision. Barring synthetic or in-studio domains, acquiring such supervision for each new target environment is highly inconvenient. To this end, we cast 3D pose learning as a self-supervised adaptation problem that aims to transfer the task knowledge from a labeled source domain to a completely unpaired target. We propose to infer *image-to-pose* via two explicit mappings viz. *image-to-latent* and *latent-to-pose* where the latter is a pre-learned decoder obtained from a prior-enforcing generative adversarial auto-encoder. Next, we introduce relation distillation as a means to align the unpaired cross-modal samples *i.e.* the unpaired target videos and unpaired 3D pose sequences. To this end, we propose a new set of non-local relations in order to characterize long-range latent pose interactions unlike general contrastive relations where positive couplings are limited to a local neighborhood structure. Further, we provide an objective way to quantify non-localness in order to select the most effective relation set. We evaluate different self-adaptation settings and demonstrate state-of-the-art 3D human pose estimation performance on standard benchmarks.[1]

## 1   Introduction

Human pose estimation systems have garnered immense attention due to their innumerable applications [55, 19, 85]. The successes of such systems are mostly driven by large-scale datasets containing images with paired 3D pose annotations [25]. Unlike 2D joint landmarks, annotating a 3D pose requires body-worn sensors or multi-camera structure-from-motion setup [71] which is challenging to install outdoors. Hence, the available datasets are either captured in constrained laboratory settings or are limited in size and diversity. Unsurprisingly, models trained on such datasets exhibit poor cross-dataset generalization. Addressing this, several approaches [11, 77, 56, 34] resort to weakly-supervised methods that rely on images with paired 2D landmark annotations. Certain methods require additional supervision such as depth [81, 11] or multi-view image pairs [66, 33]. However, they still suffer from dataset-bias due to their strong reliance on some form of paired supervision.

In this work, we digress from any form of paired supervision or auxiliary cues (multi-view or depth) thereby avoiding the curse of dataset-bias. We thus introduce a self-supervised domain adaptation framework for 3D human pose estimation (Fig. 1). In the proposed setting, we consider access to three different datasets. First, a labeled source dataset obtained from graphics-based synthetic environments such as SURREAL [79] or in-studio datasets such as Human3.6M [25]. Second, unlabeled video sequences from the target domain. Third, a set of unpaired 3D pose sequences. Following this, *image-to-pose* inference is carried out via two explicit mappings *i.e.*, *image-to-latent* CNN followed by a *latent-to-pose* network. Here, the *latent-to-pose* network is pre-learned as a decoder of a prior-enforcing generative adversarial auto-encoder [47] (AAE) in order to restrict the

---

[1]Webpage: https://sites.google.com/view/sa3dhp

pose predictions within the plausibility limits. We follow the same to pre-learn a generative motion embedding via a recurrent AAE setup which operates on the latent pose space instead of the raw pose sequences. Both pose and motion embeddings are trained solely on the unpaired 3D pose dataset. Next, we prepare *image-to-latent* CNN by supervising on the labeled source dataset. After finishing the above pre-learning steps, our prime objective is to train or adapt only the *image-to-latent* CNN so that it can perform well on the unlabeled target domain samples. In order to align the output manifold of the *image-to-latent* with the pre-learned latent pose manifold (in the absence of cross-modal pairs), one must formalize innovative ways to represent the *dark-knowledge* based on which the student (*i.e.* *image-to-latent*) output can align to that of the teacher (*i.e.* *pose-to-latent*).

Several studies [18, 20] on human cognitive development advocate that, in a self-supervised paradigm, new knowledge is acquired by relating entities based on some semantic rule. Motivated by this, we explore different ways of formalizing inter-entity relations. A relation can be characterized as lower order or higher order based on the number of entities that participate in defining a relationship tuple. For instance, in contrastive learning [57, 12], a pose space triplet relation expresses a coupling of just 3 pose entities, and is thus considered a lower order relation. However, a similar contrastive triplet defined in the hierarchical motion space (*i.e.* temporal pose sequences of fixed sequence length $T$) would couple $3T$ pose entities, thus is considered a higher order relation.

Next, we adapt the target-specific *image-to-latent* by minimizing the relational energies derived from the contrastive relations. Unlike in prior-arts [12, 51] where the output embedding is learned from scratch, we are restricted to operate on a pre-learned output pose embedding. Consequently, the model often converges to a degenerate solution exhibiting instance-level misalignment [46]. We realize that the positive coupling in contrastive relations is limited to local neighborhood structures resulting in suboptimal alignment. This motivates us to develop a new set of relations that would express positive coupling of diverse non-local relations (beyond the structural neighborhood), thereby characterizing long-range latent pose interactions in a much effective manner.

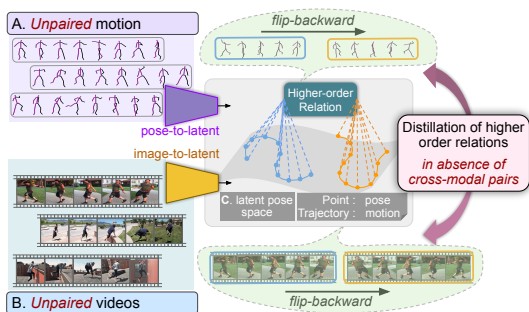

Figure 1: We align samples from unpaired pose (or motion) and unpaired images (or videos) at a shared latent pose space by distilling higher order (associating multiple instance via motion) non-local (*e.g.* `flip-backward`) relations. Relations are equivalent to a form of data-interlinking as done in knowledge-graphs.

We define tangible non-local relations separately in the pose and motion space that are categorized under a) lower order non-local relations and b) higher order non-local relations, respectively. For instance, "`pose-flip`" is a lower order non-local relation which associates anchor poses with their left-right flipped versions. Similarly, "`flip-backward`" is a higher order non-local relation which couples anchor pose-sequences with their flip-backward counterpart which is obtained via temporal reversal (backwards) of the individually flipped frames. Here, the corresponding relational energy is devised via latent space relation networks. These are essentially frozen neural networks that are trained to regress the latent embedding of the relational counterpart given latent embedding of the anchor as input. In a nutshell, the relational energies aim to preserve the equivariance of higher order spatio-temporal relations between the two modalities as a means to perform the cross-modal alignment. It turns out that, among various relations, relations coupling the most diverse non-local samples result in a better cross-modal alignment. We perform extensive experiments to validate the efficacy of our approach and demonstrate superior generalizability on samples from in-the-wild environments. We summarize our contributions as follows:

- The proposed solution for self-adapting 3D human pose model involves cross-modal alignment between the unpaired samples from the input and output modalities, via relation distillation. Highly non-local instances are associated using novel relation networks to specifically cater to the instance-level misalignment.

- We provide insights to select the most effective non-local relations. This involves quantifying non-localness of a relation as the average latent-distance between the coupled entities.

- We evaluate different self-adaptation settings and demonstrate state-of-the-art 3D human pose estimation performance against the available semi-supervised and weakly-supervised prior arts on Human3.6M [25], MPI-INF-3DHP [49], and 3DPW [80].

## 2 Related Works

Table 1 shows a comparison of the proposed setting against the prior-arts for the task of 3D human pose estimation. Our framework is equipped with self-adaptation capability while only utilizing unpaired 3D poses and synthetic 3D pose supervision.

**Knowledge distillation (KD).** The process involves transferring knowledge from a larger model to a smaller model to learn a concise knowledge representation with negligible information loss. Faster evaluation and reduced memory footprint renders distilled models more deployment friendly. More specifically, in KD [24], the smaller model (student) focuses on mimicking the output behaviour of a pre-trained larger model (teacher). Traditional techniques aim to match instance level behaviours such as matching the output activation [68], gram matrix [87], gradient [72], attention maps [88], etc. Recently, several techniques proposed to align pairwise distance [76, 63] or similarity graphs [58, 43] to look beyond the instance-level matching. Further, to capture higher order output dependencies, researchers adopt techniques [75] inspired from the advances in self-supervised contrastive learning literature [57, 4]. Essentially, a balanced distillation of both structural and instance-level knowledge leads to superior generalization [83]. However, in the proposed setting, the teacher, *pose-to-latent* and the student, *image-to-latent* do not share the same input modality. Further, the absence of paired correspondence restricts us from directly employing such distillation techniques.

**Monocular 3D human pose estimation.** There are two broad categories of models that infer 3D poses from a single image, *i.e.* a) those using model-based parametric representation [5, 8, 62, 7] and b) others that infer the 3D pose representation directly [69, 2, 6]. Models belonging to the first category map the input image to the latent parameters of a pre-defined parametric human model. This setting establishes the base to impose various pose priors through techniques such as adversarial training. Recent works [28, 61] have also aimed to build on the

Table 1: Characteristic table comparing positive and negative attributes of Ours against prior works.

| Methods | Real Paired Sup. | | | Unpaired 3D pose Sup. | Synthetic 3D pose Sup. | Self-adaptation capability |
|---|---|---|---|---|---|---|
| | 3D pose | 2D pose | Multi view | | | |
| Rhodin *et al.* [65] | ✓ | ✗ | ✓ | ✗ | ✗ | ✗ |
| Chen *et al.* [13] | ✗ | ✓ | ✓ | ✗ | ✗ | ✗ |
| Wandt *et al.* [81] | ✗ | ✓ | ✗ | ✓ | ✗ | ✗ |
| Chen *et al.* [11] | ✗ | ✓ | ✗ | ✓ | ✗ | ✗ |
| Doersch *et al.* [16] | ✗ | ✓ | ✗ | ✗ | ✓ | ✗ |
| Iqbal *et al.* [26] | ✗ | ✓ | ✓ | ✗ | ✗ | ✗ |
| Kundu *et al.* [38] | ✗ | ✓ | ✗ | ✓ | ✗ | ✗ |
| Zhang *et al.* [90] | ✗ | ✓ | ✗ | ✗ | ✗ | ✓ |
| Ours | ✗ | ✗ | ✗ | ✓ | ✓ | ✓ |

parametric body models and extend it to express finer movements such as hand gestures and facial expressions. Those belonging to the second directly map the input image to the corresponding 3D pose. These models are further categorized into a) one-stage methods [93, 73, 59, 54, 60, 86] that directly map input image to the 3D pose, and b) two-stage methods [92, 48, 53, 73] that adopt a mapping from the image to an intermediate 2D pose, followed by lifting of 2D-to-3D. In our work, the shared latent pose can be seen as a parametric form to represent plausible 3D poses.

**Domain Adaptation.** A model trained on some source data may not be a suitable choice for deployment in its nascent form to an unknown target environment. In such a scenario, domain adaptation aims to tackle this domain shift induced due to the difference in source and target data distributions by adapting the model trained on some related labeled source domain to the target domain. A number of works [37] aim to tackle the problem of domain shift induced due to the vastly different training and deployment scenarios. For the animal pose estimation task, Cao *et al.* [9] apply discriminator based discrepancy minimization technique. Zhang *et al.* [91] leverage multi-modal input such as depth and body segmentation masks. Similarly, Doersch *et al.* [16] utilize optical-flow and 2D key-points as input representations. Further, Zhang *et al.* [90] propose to use 2D landmarks, obtained from a powerful off-the-shelf 2D pose detector, as the input modality. They perform instance-level geometry-aware self supervised learning on each target for inference time 2D-to-3D lifting. These approaches rely on alternative input modalities that are least affected by domain shift unlike the raw RGB images. The proposed approach does not utilize any such auxiliary inputs thus addressing one of the most challenging adaptation settings.

## 3 Approach

In the following, we first define the basic notations and network components required to discuss the proposed training setup. Then we provide more details about the proposed self-supervised adaptation.

### 3.1 Notations and pre-learning steps

We start with listing the available datasets based on which we introduce our learning setup.

**3.1.1 Datasets.** We consider access to three different datasets as follows (refer Fig. 2). **a)** A labeled source dataset $(x^s, y^s) \in \mathcal{D}^s$, where $(x^s, y^s)$ denotes a tuple of a source image paired with its 3D pose annotation, **b)** A set of unlabeled video sequences from the target domain, $X = [x_1, x_2, .., x_T] \in \widetilde{\mathcal{X}}$ where $T$ denotes the sequence length. Here, $x_t \in \mathcal{X}$ denotes a single RGB image frame sampled from the target domain sequence $X$. And, **c)** a set of unpaired 3D pose sequences, $Y = [y_1, y_2, .., y_T] \in \widetilde{\mathcal{Y}}$. Here, a single 3D pose frame is represented as $y_t \in \mathcal{Y}$.

**3.1.2 Network components.** The *image-to-pose* inference is to be carried out via the *image-to-latent* CNN, $\mathtt{G} : \mathcal{X} \to \mathcal{Z}$ followed by a *latent-to-pose* network, $\mathtt{D_p} : \mathcal{Z} \to \mathcal{Y}$ (see Fig. 2C). We introduce two latent embeddings, i.e., pose embedding and motion embedding, as follows.

**a) The pose embedding** is denoted as $z \in \mathcal{Z}$ is constrained to follow a uniform prior distribution, $z \in [-1, 1]^{32}$. This is realized by training

Figure 2: Proposed training setup. **A** and **B**. Training pose and motion AAEs on the unpaired pose and motion data. **C.** Initializing $\mathtt{G}^s$ by supervising on labeled source data. **D.** Adapting $\mathtt{G}$ on unlabeled target videos.

an AAE (adversarial auto-encoder [47, 35, 36]) whose encoder and decoder mappings are represented as, $\mathtt{E_p} : \mathcal{Y} \to \mathcal{Z}$ and $\mathtt{D_p} : \mathcal{Z} \to \mathcal{Y}$ (Fig. 2A).

**b) The motion embedding** is denoted as $v \in \mathcal{V}$. In line with the preparation of pose embedding, the encoder and decoder mappings of the motion-AAE (see Fig. 2B) are denoted as $\mathtt{E_m} : \widetilde{\mathcal{Z}} \to \mathcal{V}$ and $\mathtt{D_m} : \mathcal{V} \to \widetilde{\mathcal{Z}}$. Here, the motion embedding is learned as a hierarchical temporal embedding of sequence of pose embeddings, $Z = [z_1, z_2, .., z_T] \in \widetilde{\mathcal{Z}}$. Moreover, the recurrent auto-encoder operates on the sequence of pose embeddings ($Z \in \widetilde{\mathcal{Z}}$) instead of the raw pose sequences ($Y \in \widetilde{\mathcal{Y}}$).

**3.1.3 Pre-learning steps.** We introduce separate domain-specific *image-to-latent* mappings for the source and target domains *i.e.*, $\mathtt{G}^s : \mathcal{X}^s \to \mathcal{Z}$ and $\mathtt{G} : \mathcal{X} \to \mathcal{Z}$ respectively. The pose and motion auto-encoders are trained using the unpaired 3D pose data, $Y \in \widetilde{\mathcal{Y}}$ and kept frozen in later training stages (see Fig. 2). Here, $\mathtt{G}^s$ is prepared by training on $\mathcal{D}^s$ using the following loss term: $||\mathtt{D_p} \circ \mathtt{G}^s(x^s) - y^s||$, where $\circ$ denotes functional composition. Given a target image-sequence (or video) $X$, its motion embedding is obtained as $\hat{v} = \mathtt{E_m} \circ \mathtt{G}(X)$. Here, $\mathtt{G}(X)$ separately processes each temporal frame of $X$ to obtain the latent pose sequence $\hat{Z}$ which is passed through $\mathtt{E_m}$ to obtain $\hat{v}$. Sec. 3.3 details the procedure to adapt $\mathtt{G}$ after initializing it from $\mathtt{G}^s$.

## 3.2 Problem formulation

The prime objective is to train or adapt the *image-to-latent* mapping, $\mathtt{G}$ to better generalize it to the unlabeled target domain samples (*i.e.* $x$ or $X$).

**a) Domain adaptation perspective.** One can see it as an unsupervised domain adaptation (DA) problem [17, 70] which aims to adapt a source trained $\mathtt{G}^s$ in order to obtain a target specific mapping $\mathtt{G}$. However, unlike segmentation or classification, 3D pose is a structured regression output. Consequently, the usual statistical or adversarial discrepancy minimization based [78, 44] DA solutions usually fail to improve the adaptation performance against the corresponding pre-adaptation baselines [16].

**b) Knowledge distillation perspective**. The problem in hand can also be perceived as a manifold alignment problem [82] where the output manifold of the *image-to-latent* $\mathtt{G}$, needs to be aligned with the pre-learned pose manifold, $\mathcal{Z}$. In knowledge distillation (KD), the same is achieved by formalizing the *dark-knowledge* based on which the output of a student network can align to that of a teacher network. However, in the proposed setting, the teacher, *pose-to-latent* $\mathtt{E_p} : \mathcal{Y} \to \mathcal{Z}$ and the student, *image-to-latent* $\mathtt{G} : \mathcal{X} \to \mathcal{Z}$ do not share the same input modality (*i.e.* image space $\mathcal{X}$ vs. pose space $\mathcal{Y}$). The most common form of distillation aims to align the instance-wise embeddings [24, 68]. However, such approaches are infeasible in the absence of cross-modal pairing *i.e.*, $(x, y) \in \mathcal{X} \times \mathcal{Y}$ in

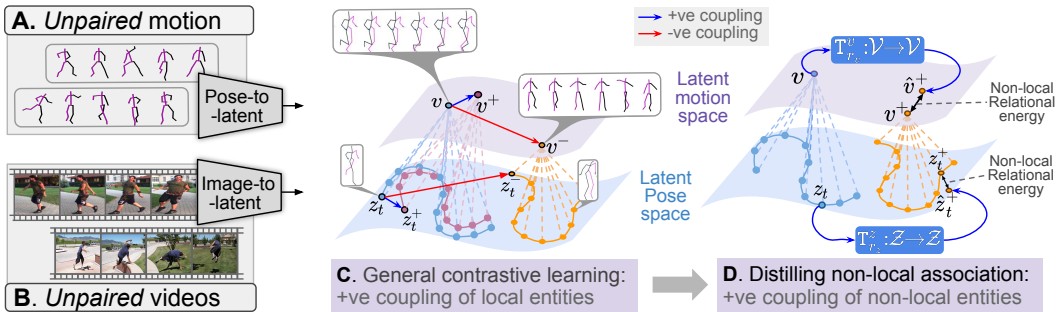

Figure 3: We align samples from unpaired pose (or motion) and unpaired images (or videos) at a shared latent pose space by distilling higher order (associating multiple instance via motion) non-local (*e.g. flip-backward*) relations. Relations are equivalent to a form data-interlinking as done in knowledge-graphs.

the unlabeled target domain. Though one can obtain pseudo $(x, y)$ pairs by forwarding the unlabeled target images through the source specific $G^s$ (via $D_p \circ G^s : \mathcal{X} \to \mathcal{Y}$), the obtained pose predictions are highly unreliable due to the input domain-shift. This motivates us to look for innovative ways to represent the *dark-knowledge* without relying on cross-modal pairs.

### 3.3 Proposed solution

We propose to represent the *dark-knowledge* by forming relational associations that aim to relate a group of intra-modal entities. For example, distilling triplet associations as used in self-supervised contrastive learning [12, 51] literature can be seen as a trivial form of representing the *dark-knowledge*.

#### 3.3.1 Distilling local neighborhood relations via contrastive learning

General contrastive learning can be applied at both latent pose space, $\mathcal{Z}$ and motion space, $\mathcal{V}$.

**a) Lower-order contrastive.** Pose space contrastive learning relies on relational associations of the form $\mathcal{C}^{\mathbf{x}} = \{(x_t, x_t^+, \{x_t^-\})$ where $(x_t, x_t^+) \to positive$ and $(x_t, \{x_t^-\}) \to negative\}$ (Fig. 3C, bottom panel). Here, $\{x_t^-\}$ represents a batch of unique instances that are in no way related to $x_t$. *positive* indicates a positive coupling of $x_t$ and $x_t^+$. Here, $x_t^+$ is obtained via a pose-invariant image space augmentation of $x_t$ *i.e.*, $x_t^+ = A(x_t)$, and $G(x_t) = G(x_t^+)$. However, in the *negative* coupling, one just knows that the corresponding poses are unrelated. Thus, $\mathcal{C}^x$ mostly focuses on characterizing the lower-order latent local neighborhood structure. Drawing inspiration from InfoNCE [57], the corresponding relational energy loss (with $\tau$ as the temperature hyperparameter) is represented as,

$$\mathcal{L}_{LCR} = -\log \sum_{(x_t, x_t^+, \{x_t^-\}) \in \mathcal{C}^x} e^{(G(x_t) \cdot G(x_t^+)/\tau)} / (e^{(G(x_t) \cdot G(x_t^+)/\tau)} + \sum_{\{x_t^-\}} e^{(G(x_t) \cdot G(x_t^-)/\tau)}) \quad (1)$$

**b) Higher-order contrastive.** Similarly, motion space contrastive characterizes higher-order local neighborhood relation *i.e.*, $\mathcal{C}^{\mathbf{X}} = \{(X, X^+, \{X^-\})$ where $(X, X^+) \to positive$ and $(X, \{X^-\}) \to negative\}$. The corresponding relational energy is denoted as $\mathcal{L}_{HCR}$. $\mathcal{L}_{HCR}$ takes the same form as of Eq 1 by replacing $G(x_t)$ with $E_m \circ G(X)$, and similarly for $X^+$ and $X^-$ (Fig. 3C, top panel).

**Higher-order vs lower-order relations.** A natural question that arises is: *why to use motion embedding when the goal task is to realize an image-to-pose mapping?* We hypothesize that formalization of higher-order pose association would enhance the expressibility of the *dark-knowledge* which is of prime interest towards realizing superior cross-modal alignment. Here, higher-order refers to relational association of a large number of pose entities. In $\mathcal{C}^X$, each individual triplet relation expresses a positive coupling of $2T$ pose entities against just 2 in $\mathcal{C}^x$. Thus, the hierarchical motion embedding facilitates a suitable platform to formalize temporally structured (temporal order must be retained) higher-order relational associations of the pose space entities (see Fig. 3C).

#### 3.3.2 Distilling non-local relations via equivariance consistency

We observe that adapting the *image-to-latent* mapper $G$ just by minimizing $\mathcal{L}_{LCR}$ and $\mathcal{L}_{HCR}$ results in poor pose estimation performance. In general self-supervised contrastive learning prior-arts, the latent embedding is learned from scratch alongside the training of *image-to-latent* mapping. However, in the proposed setting, we aim to adapt the *image-to-latent* network $G$ in order to align its output with the pre-learned pose and motion embeddings (*i.e.* the latent embeddings of frozen $\{E_p, D_p\}$ and

$\{\mathtt{E_m}, \mathtt{D_m}\}$). Consequently, we need to address a special degenerate scenario where the distillation losses completely converge even when the model exhibits severe instance-level misalignment [46].

**Local vs non-local relations.** We hypothesize that relations having non-local positive coupling, equivalent to relations in a knowledge-graph (KG), are the best way to express the instance-grounded *dark-knowledge*. Note that, the positive coupling in contrastive-based relations are limited to local neighborhood, as here, the positive counterpart of an anchor is obtained via simple pose-invariant augmentations (*i.e.* $x_t^+ = \mathtt{A}(x_t)$ in both $\mathcal{C}^x$ and $\mathcal{C}^X$). To this end, our goal is to come up with a new set of relational associations with non-local positive couplings. This aims to characterize long-range latent pose interactions, *i.e.* a positive coupling of two entities that are grounded far away in the latent pose or motion space (see Fig. 4C). To this end, we introduce relation networks.

**Relation networks.** Relation networks operating on the pose and motion embeddings are neural network mappings of the form, $\mathtt{T}_{r_z}^z : \mathcal{Z} \to \mathcal{Z}$ and $\mathtt{T}_{r_v}^v : \mathcal{V} \to \mathcal{V}$ respectively. Here, $r_z$ and $r_v$ denote independent embedding specific rule-ids for the pose and motion embeddings respectively. The equivalent rule-specific image space relation transformation operating on the image and image-sequence are represented as $\mathtt{T}_{r_z}^x : \mathcal{X} \to \mathcal{X}$, and $\mathtt{T}_{r_v}^X : \widetilde{\mathcal{X}} \to \widetilde{\mathcal{X}}$. And the same on the raw pose and motion space is represented as $\mathtt{T}_{r_z}^y : \mathcal{Y} \to \mathcal{Y}$, and $\mathtt{T}_{r_v}^Y : \widetilde{\mathcal{Y}} \to \widetilde{\mathcal{Y}}$. Note that, except $\mathtt{T}_{r_z}^z$ and $\mathtt{T}_{r_v}^v$, all other transformations are simple affine-like operations without involving neural network mappings.

**a) Distilling lower-order non-local relations.** For each target image $x_t \in \mathcal{X}$, we first construct a dataset of positive coupling represented as $\mathcal{N}_{\mathbf{r_z}}^{\mathbf{x}} = \{(x_t, x_t^+)$ where $x_t^+ = \mathtt{T}_{r_z}^x(x_t)\}$. Here, one is already aware of the corresponding pose relation, *i.e.* $y_t^+ = \mathtt{T}_{r_z}^y(y_t)$. And, the same relation in the latent pose space is expressed as $z_t^+ = \mathtt{T}_{r_z}^z(z_t)$.

Let us consider a non-local relation termed as "`pose-flip`" with rule-id $r_z = 1$. Here, $\mathcal{N}_1^x$ is obtained by pairing each target image with its flipped version obtained via a simple image-flip transformation operation, (*i.e.* $\mathtt{T}_1^x$). Note that, a similar pose space transformation would involve swapping the left side joints with that of the right (*i.e.* $\mathtt{T}_1^y$). And, the corresponding relation network $\mathtt{T}_1^z$ is a multi-layer neural network which is trained to regress $z_t^+ = \mathtt{E_p} \circ \mathtt{T}_1^y(y_t)$ while feeding $z_t = \mathtt{E_p}(y_t)$ as the input. Here, $y_t$ is sampled from the unpaired 3D pose data $\mathcal{Y}$. Next, we formalize the corresponding relational energy loss that uses the frozen relation network $\mathtt{T}_1^z$ and is represented as,

$$\mathcal{L}_1^z = \sum_{(x_t, x_t^+) \in \mathcal{N}_1^x} \|\mathtt{T}_1^z \circ \mathtt{G}(x_t) - \mathtt{G}(x_t^+)\| \tag{2}$$

**b) Distilling higher-order non-local relations.** One can define higher order non-local relations on the hierarchical motion space to devise a similar relational energy loss.

Let us consider a non-local relation termed as "`flip-backward`" with rule-id $r_v = 1$. Here, $\mathcal{N}_1^{\mathbf{X}} = \{(X, X^+)$ where $X^+ = \mathtt{T}_1^X(X)\}$ is obtained by pairing each target sequence $X$ with its `flip-backward` counterpart which is obtained via temporal reversal (backwards) of the image-flipped frames. Accordingly, the corresponding relation network $\mathtt{T}_1^v$ is a multi-layer neural network trained to regress $v^+ = \mathtt{E_m} \circ \mathtt{E_p} \circ \mathtt{T}_1^Y(Y)$ while feeding $v = \mathtt{E_m} \circ \mathtt{E_p}(Y)$ as the input, where $Y$ is sampled from the unpaired 3D pose sequences $\widetilde{\mathcal{Y}}$. And, the corresponding relational energy loss is represented as,

$$\mathcal{L}_1^v = \sum_{(X, X^+) \in \mathcal{N}_1^X} \|\mathtt{T}_1^v \circ \mathtt{G}(X) - \mathtt{G}(X^+)\| \tag{3}$$

Finally, cross-modal alignment is achieved by simultaneously minimizing all the relational energies, *i.e.*

$$\mathcal{L} = \mathcal{L}_{CR} + \mathcal{L}_{NLR}; \text{ where } \mathcal{L}_{CR} = \mathcal{L}_{LCR} + \mathcal{L}_{HCR} \text{ and } \mathcal{L}_{NLR} = \sum_{r_z} \mathcal{L}_{r_z}^z + \sum_{r_v} \mathcal{L}_{r_v}^v \tag{4}$$

Next, we discuss certain insights which can help us to formalize effective non-local relations.

### 3.3.3 What makes non-local relations more effective?

In the latent pose space, we conceptualize two different relation networks viz. a) `pose-flip`, $\mathtt{T}_1^z$ and b) `in-plane-rotation`, $\mathtt{T}_2^z$. The corresponding relational energies are denoted as $\mathcal{L}_1^z$ and $\mathcal{L}_2^z$ respectively. Though minimizing $\mathcal{L}_1^z$ yields a substantial improvement over *Ours-LL* (just minimizing $\mathcal{L}_{CR}$ without $\mathcal{L}_{NLR}$), jointly minimizing both $\mathcal{L}_1^z$ and $\mathcal{L}_2^z$ does not yield much improvement on the final pose estimation performance. However, the combination of flip and rotation in a single relation yields a considerable improvement in performance. We call this rule "`flip+inplane-`$\theta$", with rule-id $r_z = 3$. We also notice improvement in performance when using in-plane rotation used in conjunction with the higher order relation "`flip-backward`" (rotation of individual flipped frames). This rule

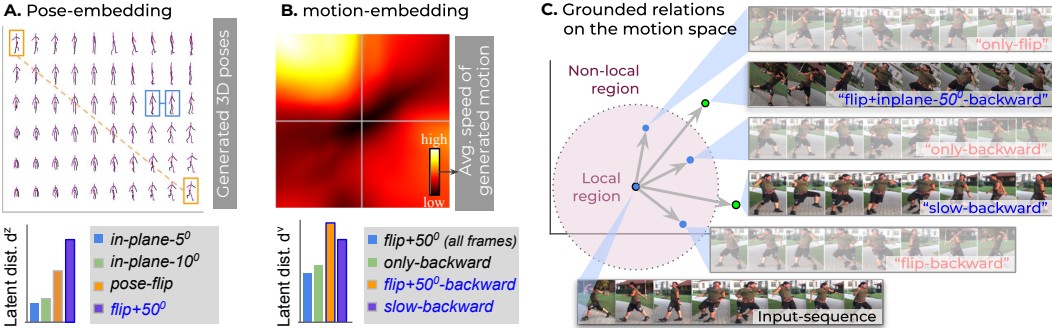

Figure 4: Embedding visualizations. **A.** Grid-interpolation of pose embedding (`pose-flip`: orange boxes; `in-plane-5°`: blue boxes) with bar-plot showing the avg. latent-distance. **B.** Grid-interpolation of speed of the generated motions with latent-distance bar-plot comparing the non-local motion relations. **C.** 2D PCA projected motion embedding points with glances of the anchor video (middle-point) and the corresponding transformed videos obtained by operating over various motion-related non-local relation candidates. Note that, only the relations in blue are selected because of their high non-localness indicated by the latent-distances $d^z$ and $d^v$.

is termed "`flip+inplane-θ-backward`", with rule-id $r_v = 2$. We also introduce a higher-order non-local relation "`slow-backward`", with rule-id $r_v = 3$. The relational pairs for `slow-backward` are constructed by sampling the original sequence at 15 FPS, to enable a temporally slow 30 FPS slow backward video taken from the middle region (see Fig. 4C).

**Quantifying non-localness via latent-distance.** We hypothesize that, "*relations coupling diverse samples (long-range interactions) lead to better cross-modal alignment*". To evaluate this hypothesis, we define a distance metric to apprehend the extent of non-localness of individual relations which is termed as *latent-distance*. Here, the *latent-distance* $\mathbf{d}_1^z = \text{mean}_{(z_t, z_t^+) \in \mathcal{N}_1^z} \| z_t - z_t^+ \|$ measures the average distance between relation pairs ($z_t = \mathbb{E}_p(y_t), z_t^+ = \mathbb{E}_p \circ \mathsf{T}_1^y(y_t)$), for relation-id $r_z = 1$, where $y(y_t)$ is sampled from the unpaired 3D pose data $\mathcal{Y}$. In Fig. 4A, the bar-plot comparing the *latent-distance* for various relations clearly highlight the gap in the degree of non-localness thus justify-

---

**A. Pre-learning steps** (training on source data)
1) Train $\mathbb{E}_p$ and $\mathbb{D}_p$ on unpaired 3D poses, $y \in \mathcal{Y}$
2) Train $\mathbb{E}_m$ and $\mathbb{D}_m$ on unpaired motion, $Y \in \widetilde{\mathcal{Y}}$
3) Train $\mathbb{G}^s$ on labeled source dataset, $(x^s, y^s) \in \mathcal{D}^s$
4) Train $\mathsf{T}_3^z$ for "`flip+inplane-θ`" on $\mathcal{Z}$ using $y \in \mathcal{Y}$
5) Train $\mathsf{T}_2^v$ for "`flip+inplane-θ-backward`" on $\mathcal{V}$ using $Y \in \widetilde{\mathcal{Y}}$
6) Train $\mathsf{T}_3^v$ for "`slow-backward`" on $\mathcal{V}$ using $Y \in \widetilde{\mathcal{Y}}$

**B. Unsupervised target adaptation**
7) After initializing $\mathbb{G}$ from $\mathbb{G}^s$, train $\mathbb{G}$ by minimizing the local relational energy, $\mathcal{L}_{CR}$ alongside minimizing the non-local relations $\mathcal{L}_3^z$, $\mathcal{L}_2^v$ and $\mathcal{L}_3^v$.

**Algorithm 1:** Overview of the optimization steps. Note that, $\mathcal{L}_3^z$, $\mathcal{L}_2^v$ and $\mathcal{L}_3^v$ rely on frozen $\mathsf{T}_3^z$, $\mathsf{T}_2^v$ and $\mathsf{T}_3^v$ to distill the embedded relational *dark-knowledge*.

---

ing the aforementioned observation regarding the performance gap. We perform a similar metric analysis to select the best set of relations defined on the motion space $\mathcal{V}$ (see Fig. 4B and 4C).

**Finalizing the non-local relations.** At the end, we take up a single pose relation, *i.e.*, `flip+inplane-θ` and two diverse motion relations, *i.e.*, `flip+inplane-θ-backward` and `slow-backward` to realize our final cross-modal alignment framework (Fig. 4). With $\mathcal{L}_3^z$ being the relational energy specific to the pose space relation `flip+inplane-θ`, we denote $\mathcal{L}_2^v$ and $\mathcal{L}_3^v$ as the relational energies specific to the motion relations `flip+inplane-θ-backward` and `slow-backward` respectively. Please refer to Supplementary for an ablation on the effect of changing the angle $\theta$.

**Overview of the optimization steps.** Algorithm 1 summarizes the training as a step-by-step learning process. Here, each step uses frozen models obtained from the previous step. After initializing $\mathbb{G}$ from $\mathbb{G}^s$ only a minimal set of network parameters (*Res-3* block of ResNet-50) are updated to learn target-specific mapping. This can be see as a regularization for the unsupervised adaptation training. We freeze $\mathbb{D}_p$ while training $\mathbb{G}$ as it is crucial to regularize the unsupervised adaptation to avoid degenerate solutions (or mode-collapse). Updating $\mathbb{D}_p$ while training $\mathbb{G}$ requires us to impose an additional adversarial discriminator loss (as used in HMR [29]) in order to uphold its ability to decode plausible pose predictions. Note that, updating $\mathbb{D}_p$ would update the manifold structure of

Table 2: Comparison of 3D pose estimation results on Human3.6M dataset. *Full (3D) supervision* denotes using GT 3D supervision for training. *Semi-Sup.* denotes supervised training only on subject S1. *Unsup.* refers to training on unpaired Human3.6M samples (unlabeled). We achieve SOTA results on both semi-supervised and unsup. training methods. Our unsupervised training outperforms the SOTA even in presence of huge domain gap between the SURREAL and H3.6M datasets. (MV) denotes usage of multi-view supervision.

| Training | Methods | PA-MPJPE ↓ | MPJPE ↓ |
|---|---|---|---|
| Full (3D) Sup. | Chen *et al.* [10] | 82.7 | - |
| | Martinez *et al.* [48] | 47.7 | - |
| | Li *et al.* [41] | 38.0 | - |
| | Xu *et al.* [84] | 36.2 | 45.6 |
| | Chen *et al.* [14] | 32.7 | 47.3 |
| Semi-sup. (sup. on S1) | Mitra *et al.* [52] | 90.8 | 120.9 |
| | Li *et al.* [42] | 66.5 | 88.8 |
| | Rhodin *et al.* [65] | 65.1 | - |
| | Kocabas *et al.* [33] | 60.2 | - |
| | Iqbal *et al.* [26]$^{(MV)}$ | 51.4 | 62.8 |
| | *Ours(S→H, Semi)* | **48.2** | **57.6** |
| Unsup. | Kundu et al. [39] | 99.2 | - |
| | Kundu *et al.* [40] | 89.4 | - |
| | *Ours(S→H)* | **86.2** | 97.8 |

Table 3: Quantitative results for 3D pose estimation on 3DHP under three different supervision settings. *Full (3D) supervision* denotes using GT 3D supervision on 3DHP during training. *Unsupervised Adaptation* refers to training on labeled Human3.6M and adapting to unlabeled 3DHP. *Direct Transfer* denotes training on labeled source dataset, adapting to unlabeled web-dataset and evaluating on unseen 3DHP dataset. * denotes the implementation taken from [89]. (MV) denotes using multiview data.

| Training | Methods | PCK ↑ | AUC ↑ | PA-MPJPE ↓ |
|---|---|---|---|---|
| Full (3D) Sup. | Rogez *et al.* [67] | 59.6 | 27.6 | 158.4 |
| | Mehta *et al.* [50] | 76.6 | 40.4 | 124.7 |
| | Kocabas *et al.* [33] | 77.5 | - | - |
| | Iqbal *et al.* [26] | 83.0 | - | - |
| Unsup. Adapt. | Iqbal *et al.* [26]$^{(MV)}$ | 76.5 | - | 122.4 |
| | Kundu *et al.* [40] | 79.2 | 43.4 | 99.2 |
| | Wandt *et al.* [81]* | 81.6 | 47 | 95.4 |
| | Kundu *et al.* [39] | 83.2 | 58.7 | 97.6 |
| | Zhao *et al.* [92]* | 86.0 | 46.7 | 96.8 |
| | *Ours(H→M)* | **89.8** | **59.3** | **79.6** |
| Direct Transfer | Chen *et al.* [11] | 64.3 | 31.6 | - |
| | Kundu *et al.* [40] | 76.5 | 39.8 | 115.3 |
| | Li *et al.* [41] | 81.2 | 46.1 | - |
| | Kundu *et al.* [39] | 82.1 | 56.3 | 103.8 |
| | *Ours(S→W)* | 79.1 | 43.4 | 114.9 |
| | *Ours(SH→W)* | **85.5** | **60.7** | **94.1** |

the embedding space. Further, as the relation networks ($T_3^z$, $T_2^v$, $T_3^v$) operate on the frozen latent embeddings, the pre-learned relations networks (used to define the relation distillation objective) would no longer be useful and are required to be updated alongside. Such unconstrained optimization greatly destabilizes the self-adaptive process (in the absence of cross-modal pairs) and degrades the performance significantly. Note that the proposed self-adaptation step does not involve any complex adversarial loss component, which greatly simplifies the training procedure.

## 4    Experiments

We demonstrate effectiveness of the proposed framework by evaluating it on a variety of cross-dataset adaptation settings.

**Implementation details.** The backbone of $G^s$ constitutes of an ImageNet initialized ResNet-50 [23] (till *Res-4F*) followed by a series of fully-connected (FC) layers to obtain the latent pose representation, $z \in \mathbb{R}^{32}$. The pose auto-encoder, $\{E_p, D_p\}$ are FC networks operating on the 3D pose $y$ (17 joints). The motion auto-encoder, $\{E_m, D_m\}$ is composed of bidirectional LSTMs [22] with 128 hidden units operating on a fixed sequence length of 30 (30 FPS) where the intermediate motion-embedding, $v \in \mathbb{R}^{128}$. The relation networks constitute of simple FC layers. We associate separate Adam optimizer [32] to each relational energy term which are optimized in alternate training iterations. The adaptation is performed on an Nvidia V100 GPU with each batch containing 8 videos each of frame-length 30 (see Suppl. for more details).

**Datasets.** We use the CMU-MoCap [1] dataset as the sample set for unpaired 3D poses $\mathcal{Y}$ and unpaired pose sequences $\widetilde{\mathcal{Y}}$. We use the synthetic SURREAL (**S**) dataset [79] as one of the source datasets. The sample set for the unpaired videos $\widetilde{\mathcal{X}}$ constitutes of single-person action videos (dance forms, sports, etc.) collected from the Sports-1M dataset [31]. The raw video frames are forwarded through a person-detector [64] to obtain the person-focused image sequences. We name it as *web-dataset* (**W**). For a fair evaluation, we use the standard, in-studio Human3.6M (**H**) dataset [25] as both source or target domain, in different problem settings. MPI-INF-3DHP (**M**)  [49] is used as an unlabeled target domain to evaluate cross-studio adaptation. Further, 3DPW [80], and LSP [27] datasets are used to evaluate our cross-dataset generalizability, without involving these during training.

### 4.1    Adaptation settings

We introduce the following adaptation settings with various source and target dataset selections.

a) **Ours(S→H)** We use labeled synthetic SURREAL (S) as the source while unlabeled Human3.6M (H) acts as the target without involving the web-dataset (W). The resultant model is tuned to work well only for the specific in-studio environment, thus may fail to generalize for in-the-wild data.

b) **Ours(H→M)** Here, labeled Human3.6M is used as the only source domain while MPI-INF-3DHP (M) samples are used as the unlabeled target. This evaluates our cross-studio adaptation performance.

c) **Ours(S→W)** Here, labeled SURREAL (S) is used as the only source domain while the unlabeled target videos are extracted from the web-dataset (W). The resultant model is evaluated for cross-dataset generalization on 3DHP and 3DPW (see Table 3 and Table 4). Note that, web-dataset is the most challenging in-the-wild dataset with substantial diversity in pose, apparel, and backgrounds.

d) **Ours(SH→W)** Here, the source domain is a combination of labeled SURREAL (S) and Human3.6M (H) datasets while the unlabeled web-dataset (W) acts as the target. This setting enjoys the advantage of strong source supervision on natural but in-studio Human3.6M alongside SURREAL.

### 4.2 Evaluation on benchmark datasets

In this section, we evaluate the proposed approach on the standard benchmark datasets. MPJPE and PA-MPJPE [25] denote standard mean per joint position error metric computed before and after Procrustes Alignment [21]. Following prior arts [49], we report PCK and AUC, i.e. percentage of correct keypoints and area under the curve respectively for the 3DHP dataset.

Table 4: Quantitative results comparing our approach against prior-arts for 3D pose estimation on 3DPW. *Full (3D) supervision* denotes using ground-truth 3D supervision on 3DPW for training. *Direct Transfer* denotes training on a variety of labeled source dataset and directly evaluating the resultant model on unseen 3DPW test set. We achieve state-of-the-art result upon using both Human3.6M (H) and SURREAL (S) as the source and the web-data as the target. [+] denotes number taken from [30]. * denotes using parametric mesh models.

| Training | Methods | PA-MPJPE $\downarrow$ |
|---|---|---|
| Full (3D) Supervision | Arnab et al. [3]* | 77.2 |
| | Sun et al. [74]* | 69.5 |
| Direct Transfer | Martinez et al. [48][+] | 157.0 |
| | Dabral et al. [15][+] | 92.3 |
| | Kanazawa et al. [30]* | 80.1 |
| | Doersch et al. [16]* | 82.4 |
| | Kanazawa et al. [29]*[+] | 76.7 |
| | *Ours(S→W)* | 79.3 |
| | *Ours(SH→W)* | **72.1** |

**a) Evaluation on Human3.6M.** Table 2 lists a comparison of *Ours(S→H)* against the prior arts. Under unsupervised training setup, *Ours(S→H)* outperforms the prior state-of-the-art (SOTA) by a significant margin. We attribute this improvement to our attempt to utilize domain randomization via SURREAL and the proposed cross-dataset alignment procedure. Under semi-supervised training, the target adaptation is supervised on a small subset of labeled samples (*i.e.* subject S1 in Human3.6M) alongside unsupervised alignment on the rest. The resultant model, *Ours(S→H, Semi)* also outperforms the prior semi-supervised works even in the absence of additional multi-view supervision as used in some of the prior-arts.

**b) Evaluation on 3DHP.** Table 3 shows a detailed quantitative comparison of three of our adaptation variants; *Ours(H→M)*, *Ours(S→W)*, and *Ours(SH→W)*, under two training setups. *Ours(SH→W)* achieves state-of-the-art unseen transfer performance validating superior generalizability of the proposed framework. *Ours(SH→W)* uses in-studio but real Human3.6M as an additional labeled source data alongside SURREAL, thus reducing the domain gap when compared against *Ours(S→W)*.

**c) Evaluation on 3DPW.** Table 4 reports a detailed comparison of our variants against the prior-arts on the in-the-wild 3DPW dataset. All the methods under *direct transfer* do not use 3DPW samples even for any supervised or unsupervised training. A lower PA-MPJPE for *Ours(SH→W)* clearly highlight our superior cross-dataset generalizability against the prior approaches which also utilize additional information such as the SMPL mesh model [45].

Table 5: Ablation study on Human3.6M dataset (*i.e. Ours(S→H)*). Starting from the baseline (inference through the SURREAL trained model), usage of different relational energies results in a substantial improvement in the adaptation performance.

| Ablation | Modules Involved | MPJPE $\downarrow$ |
|---|---|---|
| Source-only | $G, D_P$ | 209.6 |
| $+\mathcal{L}_{LCR}$ | $G, D_P$ | 193.4 |
| $+\mathcal{L}_{HCR}$ | $+E_m$ | 172.1 |
| $+\mathcal{L}_3^z$ | $+T_3^z$ | 139.7 |
| $+\mathcal{L}_2^v$ | $+T_2^v$ | 91.8 |
| $+\mathcal{L}_3^v$ | $+T_3^v$ | 86.2 |

**d) Ablation study on Human3.6M.** Table 5 shows an ablative analysis highlighting the effectiveness of individual relational energies towards realizing a better unsupervised alignment. As compared

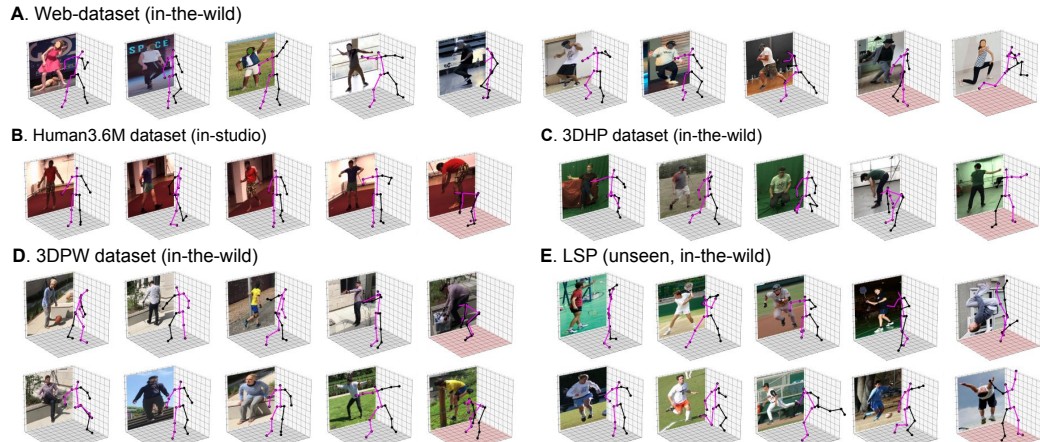

**A**. Web-dataset (in-the-wild)

**B**. Human3.6M dataset (in-studio)

**C**. 3DHP dataset (in-the-wild)

**D**. 3DPW dataset (in-the-wild)

**E**. LSP (unseen, in-the-wild)

Figure 5: Predictions obtained via *Ours(SH→W)* generalize well to both in-studio and in-the-wild datasets. The failure cases (rare poses or poses with inter-limb occlusion) are highlighted by results on red bases.

to the source-only baseline (first row of Table 5), distillation of contrastive relations in $\mathcal{L}_{\text{CR}}$ yields the first stage of improvement. Minimizing the non-local pose space relational energy $\mathcal{L}_3^z$ results in further improvement by effectively distilling an instance-grounded *dark-knowledge*. Additionally, distilling higher order non-local motion relations (*i.e.* $\mathcal{L}_2^v$ and $\mathcal{L}_3^v$) yields a substantial improvement by effectively attenuating the instance-level misalignment.

**Qualitative evaluation and limitations.** Fig. 5 illustrates the generaliability of *Ours(SH→W)* proposed approach under diverse pose, apparel, and background variations. Results with red base show some of the failure cases. The model is restricted from predicting implausible 3D pose outcomes as these are inferred through the prior-enforcing generative pose decoder. However, the model fails under certain drastic scenarios such as high background clutter, multi-level body-part occlusion, and rare athletic poses. Other limitations include body-truncation scenarios *i.e.*, scenarios where certain body-parts are either occluded by external objects or are outside the image frame. In such cases there are multiple plausible 3D pose outcomes, thus asking for a future exploration of probabilistic pose estimation modeling. Taking a different direction, one can explore innovative ways to utilize inputs from auxiliary modalities (such as depth, foreground segmentation, etc.) as and when available from the deployed target environment (see Suppl. for more details).

**Negative societal impacts.** While we do not foresee our framework causing any direct negative societal impact, it may be leveraged to create malicious applications utilizing human tracking and action recognition. Estimating a human pose does not require any identity information. However, methods such as gait recognition can be indirectly used to identify personal attributes thereby raising privacy concerns. We urge the readers to make use of the work detailed responsibly, limiting the usage to legal use-cases.

## 5 Conclusion

We presented a cross-modal alignment technique to align the learned representations from two diverse modalities. Our unsupervised technique allows adaptation to the wildest unlabeled samples gathered from web while initializing the base model on substantially diverse but unrealistic SURREAL. We analyzed the importance of higher order, non-local relations in expressing the rich instance-grounded dark knowledge as required to attenuate the instance-level misalignment. In future, we plan to extend our framework for multi-modal alignment in presence of additional input modalities.

## Acknowledgments and Disclosure of Funding

This work was supported by a Google Ph.D. Fellowship (Jogendra) and a project grant from MeitY (No.4(16)/2019-ITEA), Govt. of India.

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
