# Supplementary: Non-Local Latent Relation Distillation for Self-Adaptive 3D Human Pose Estimation

The supplementary material is organized as follows:

Table 1: Notation table. The numbers in parentheses denote the dimensions of the corresponding input. All video and motion samples are of sequence length 30. We use 17 joint 3D poses.

| | Symbol | Description |
|---|---|---|
| **Inputs** | $(x^s, y^s) \in \mathcal{D}^s$ | Labeled source dataset |
| | $y_t \in \mathcal{Y}$ | Unpaired 3D pose dataset ($17 \times 3$) |
| | $Y \in \widetilde{\mathcal{Y}}$ | Unpaired 3D motion dataset ($30 \times 17 \times 3$) |
| | $x_t \in \mathcal{X}$ | Unpaired target domain images ($224 \times 224 \times 3$) |
| | $X \in \widetilde{\mathcal{X}}$ | Unpaired target domain videos ($30 \times 224 \times 224 \times 3$) |
| **Network components** | $\{\texttt{E}_\texttt{p}, \texttt{D}_\texttt{p}\}$ | AAE for pose embedding, $\mathcal{Y} \rightarrow \mathcal{Z} \rightarrow \mathcal{Y}$ |
| | $\{\texttt{E}_\texttt{m}, \texttt{D}_\texttt{m}\}$ | AAE for motion embedding, $\widetilde{\mathcal{Z}} \rightarrow \mathcal{V} \rightarrow \widetilde{\mathcal{Z}}$ |
| | $\texttt{G}^\texttt{s} : \mathcal{X}^s \rightarrow \mathcal{Z}$ | Source-specific image-to-latent mapping |
| | $\texttt{G} : \mathcal{X} \rightarrow \mathcal{Z}$ | Target-specific image-to-latent mapping |
| | $\texttt{T}_1^z : \mathcal{Z} \rightarrow \mathcal{Z}$ | "`pose-flip`" relation network on pose |
| | $\texttt{T}_1^v : \mathcal{V} \rightarrow \mathcal{V}$ | "`flip-backward`" relation network on motion |
| | $\texttt{T}_2^v : \mathcal{V} \rightarrow \mathcal{V}$ | "`slow-backward`" relation network on motion |
| **Others** | $z \in \mathcal{Z}$ | Constrained pose space, $z \in [-1, 1]^{32}$ |
| | $v \in \mathcal{V}$ | Constrained motion space, $v \in [-1, 1]^{128}$ |

## 1 Data preparation

We use the CMU-MoCap [1] dataset to prepare sample set for unpaired 3D poses $y$ and unpaired pose sequences $Y$. The sample set for the unpaired videos $\widetilde{\mathcal{X}}$ constitutes of single-person action videos collected from Sports-1M [8]. As compared to the available in-studio datasets (such as Human3.6M and MPI-INF-3DHP) the web-dataset covers a wide diversity in apparel style, background variation, action style, etc. The raw video frames are forwarded through a person-detector [15] to obtain the person-focused image sequences. Note that, the detector pruned video sequences may not have a smooth pixel transition. However, it retains the smooth pose transition at the view-variant root-relative system. In our work, the shared latent pose can be seen as a parametric form to represent plausible 3D poses. And, the *image-to-latent* model is trained to regress the latent pose parameters with latent being an intermediate 3D pose representation.

## 2 Notations

Most of the notations used in this paper are summarized in Table 1. In the first part, we list the notations related to the datasets. The second part lists the network components as mapping functions from one space to another.

## 3 Overall training algorithm

Algorithm 1 shows the overall training algorithm under two subheadings *i.e.*, a) Pre-learning steps and b) Unsupervised alignment. Notice the chronology, as each step uses frozen models obtained from previous steps. In L10 of Algorithm 1, only the *Res-3* block parameters of G are updated for the final target adaptation. This greatly regularizes the unsupervised alignment, thereby avoiding convergence to degenerate solutions.

## 4 Ablation Experiment for rotation angle $\theta$ for inplane-$\theta$

An ablation with increasing in-plane rotation angles strongly supports our key hypothesis. Table 2 reports

---

**Algorithm 1** Overall training algorithm.

1: **require:** Unpaired motion samples $Y$, Paired source dataset $\mathcal{D}^s$, Unpaired target videos $X$.

   **A. Pre-learning steps.** *(Training on source data)*

2: **Train** $\{\mathtt{E_p}, \mathtt{D_p}\}$: AAE training on unpaired 3D poses $y \in \mathcal{Y}$ (adv. and pose reconstruction loss)

3: **Train**$\{\mathtt{E_m}, \mathtt{D_m}\}$: AE training on unpaired motions $Y \in \widetilde{\mathcal{Y}}$ (motion reconstruction loss)

4: **Train** $\mathtt{G}^s$: Training the base *image-to-latent* by minimizing $\|\mathtt{D_p} \circ \mathtt{G}^s(x^s) - y^s\|$.

5: **Train** $\mathtt{T}_3^z$: Training the `flip+inplane-`$\theta$ network on $(y, y^+)$ by $\min_{\theta_{\mathtt{T}_3^z}} \|\mathtt{T}_3^z(\mathtt{E_p}(y)) - \mathtt{E_p}(y^+)\|$.

6: **Train** $\mathtt{T}_2^v$: Training the `flip+inplane-`$\theta$`-backward` network by $\min_{\theta_{\mathtt{T}_2^v}} \|\mathtt{T}_2^v(\mathtt{E_m}(\mathtt{E_p}(Y))) - \mathtt{E_m}(\mathtt{E_p}(Y^+))\|$.

7: **Train** $\mathtt{T}_3^v$: Training the `slow-backward` network by $\min_{\theta_{\mathtt{T}_3^v}} \|\mathtt{T}_3^v(\mathtt{E_m}(\mathtt{E_p}(Y))) - \mathtt{E_m}(\mathtt{E_p}(Y^+))\|$.

   **B. Unsupervised alignment** *(Unsup. target adaptation)*

8: **while** the training has not converged **do**

9: $\quad X \leftarrow$ minibatch from unpaired target videos $\widetilde{\mathcal{X}}$

10: $\quad$ **update** trainable params of G by minimizing $\mathcal{L}_{CR}$, $\mathcal{L}_3^z$, $\mathcal{L}_2^v$, and $\mathcal{L}_3^v$ in separate *Adam* optimizers (others kept frozen).

11: **end while**

---

MPJPE (lower is better) with inplane-$\theta$ (2nd row) and `flip+inplane-`$\theta$`-backwards` (3rd row) as the lower-order (pose space) and higher-order (motion space) relations respectively (in the settings of Table 5). In both cases, $\theta$ is varied as 10, 25, 50, 75, 100, 125. In the last column, we compare the results with "`flip`" and "`flip-backwards`" in the 2nd and 3rd rows respectively.

**Observation** - We see an increase in adaptation performance with an increase in in-plane rotation for $\theta = 10, 25, 50$. However, with a further increase in $\theta$ (i.e., $75, 100, 125$) the performance seems to be saturating at a degraded level. The prime reason behind this behavior is attributed to the fact that pose samples with $\theta > 50$ are quite rare (fall in the low probability region of the latent pose space). For example, images depicting poses with the spine parallel to the ground or headstand are very rare. However, the probability of encountering images with a flipped pose is quite high, and both the original and flipped poses fall in the high probability region of the latent pose space.

**With flip+inplane** - A non-local relation with a combination of `flip` and a suitable In-Plane rotation, i.e., `flip+inplane-`$50°$ yields the best performance (in both lower and higher-order cases) beyond using just flip or In-Plane rotation.

**In summary,** the hypothesis is valid (and more effective) as long as both the relational association candidates fall in the high-probability regions of the latent pose space.

## 5 Network architecture

The *image-to-latent* model $\mathtt{G}^s$ constitutes of an ImageNet initialized ResNet-50 [6] (till *Res-4F*) followed by a series of convolution and fully-connected (FC) layers to obtain the latent pose represen-

Table 2: MPJPE when varying InPlane Rotation in settings of Table 5

| In-Plane-$\theta$ | $10°$ | $25°$ | $50°$ | $75°$ | $100°$ | $125°$ | **Flip** | **Flip+In-Plane-**$50°$ |
|---|---|---|---|---|---|---|---|---|
| Lower-order | 179.1 | 170.3 | 154.4 | 163.9 | 159.2 | 158.4 | 141.2 | **139.7** |
| Higher-order (+backwards) | 130.5 | 123.2 | 110.1 | 115.6 | 114.1 | 115.4 | 95.6 | **91.8** |

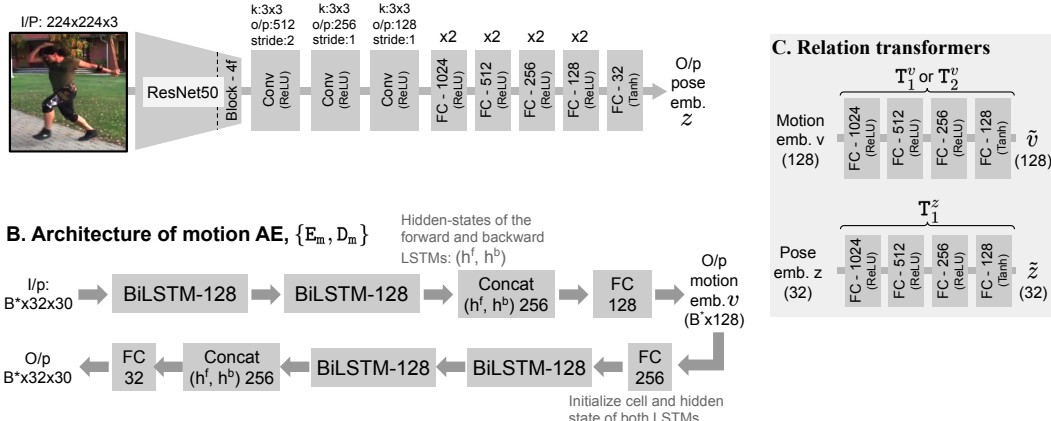

**A. Architecture of the *Image-to-latent* model**

**B. Architecture of motion AE, $\{E_m, D_m\}$**

**C. Relation transformers**

Figure 1: Detailed architecture of the network components. **A.** The *image-to-latent* model consists of the *ResNet50* backbone followed by three *conv* layers and four blocks of fully connected (FC) layers. x2 depicts 2 FCs contained in the block. **B.** The motion autoencoder is composed of two stacked BiLSTMs for the encoder and the same for the decoder. **C.** Relation networks consist of four FCs.

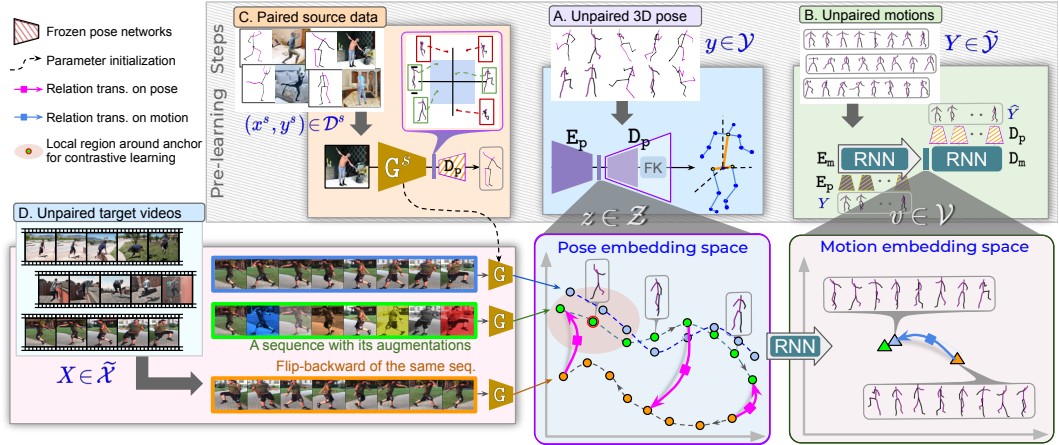

Figure 2: Framework overview. **A.** Learning pose embedding on unpaired 3D poses. **B.** Learning motion embedding on unpaired motions. **C.** Source pretraining of the image-to-latent. **D.** Adaptation to unpaired in-the-wild videos via cross-modal alignment using a) contrastive learning, b) lower-order non-local relation distillation on $\mathcal{Z}$, and c) higher-order non-local relation distillation on $\mathcal{V}$.

tation, $z \in \mathbb{R}^{32}$ (see Fig. 1A). The pose auto-encoder, $\{E_p, D_p\}$ are FC networks operating on the 3D pose $y$ (17 joint 3D coordinates). We employ the pose auto-encoder architecture as used in Kundu *et al.* [9].

The motion auto-encoder, $\{E_m, D_m\}$ is composed of bidirectional LSTMs [5] with 128 hidden units operating on a fixed sequence length of 30 (see Fig. 1B). The encoder operates on the sequence of pose embedding which is obtained by passing the pose sequence through the frozen pose encoder. Similarly, the decoder outputs a sequence of pose embedding which is forwarded through the frozen pose decoder to realize the reconstructed motion. At the encoder side, the concatenated final hidden state of both forward and backward LSTMs are forwarded through a FC layer with *tanh* non-linearity to obtain the motion embedding $v \in \mathbb{R}^{128}$. Similarly, at the decoder side, the output sequence of pose embedding is obtained via a FC layer.

The relation networks constitute of simple fully-connected layers as shown in Fig. 1C. Here, the motion relation networks are mappings defined from one instance to another in the same motion space, *i.e.*, $T_1^v : \mathcal{V} \to \mathcal{V}$, $T_2^v : \mathcal{V} \to \mathcal{V}$ and $T_3^v : \mathcal{V} \to \mathcal{V}$. Similarly, the pose transformations are expressed

**A. ($X$, $X^+$) pairs for non-local relational energy**

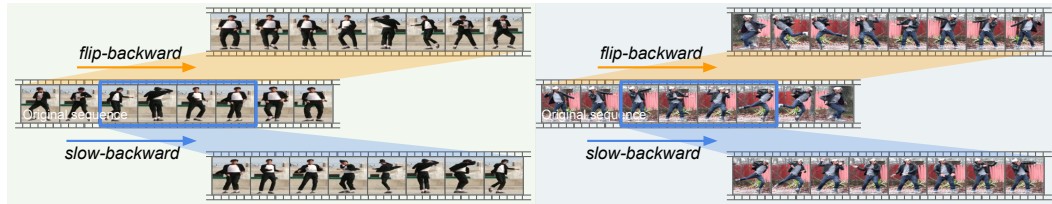

**B. ($Y$, $Y^+$) pairs to train the relational transformers**

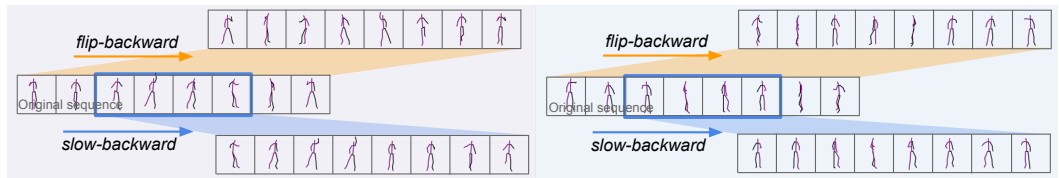

Figure 3: Example relation pairs. **A.** Relation pair examples, for `flip-backward` and `slow-backward`, constructed in the video modality, $\widetilde{\mathcal{X}}$. Note that, to construct `slow-backward` pairs, the original sequence is sampled at 15 FPS to enable a 30 FPS temporally slow video taken from the mid region (notice the blue box). **B.** Relation pair examples constructed in the motion modality, $\widetilde{\mathcal{Y}}$. Note that, we do not have access to any cross-modal correspondence information.

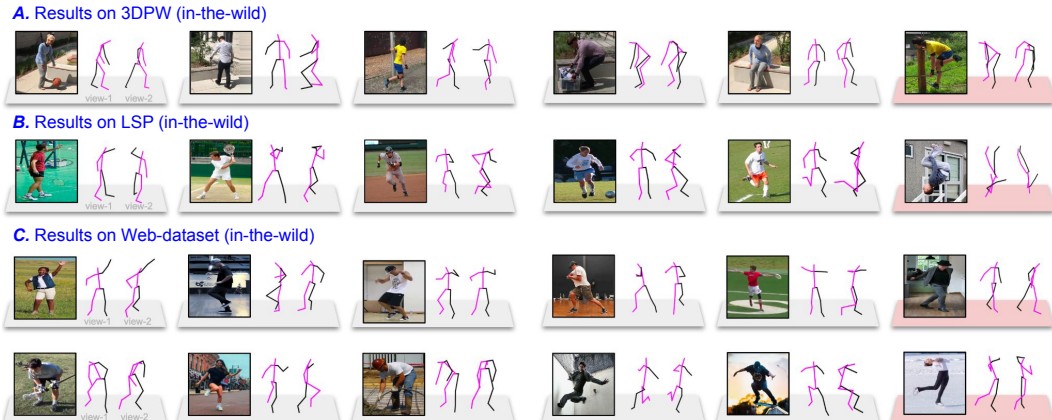

Figure 4: Qualitative analysis. The 3D pose estimation results are shown on 3DPW, LSP, and Web-datasets, with an alternate view. Predictions shown are obtained via *Ours(SH→W)*. The failure cases (rare poses, inter-limb occlusion, and high foreground-background clutter) are highlighted by results with red bases.

as, $\mathtt{T}_1^z : \mathcal{Z} \to \mathcal{Z}$, $\mathtt{T}_2^z : \mathcal{Z} \to \mathcal{Z}$ and $\mathtt{T}_3^z : \mathcal{Z} \to \mathcal{Z}$. Fig. 3 example relation pairs in the video and motion modalities.

Below, we elaborate more details of the pre-learning modules (see Fig. 2 for an overview).

**a) Pose embedding.** Human 3D pose follows a complex structural articulation constrained by the kinematic plausibility limits [3]. In this work, the motivation to learn a pose embedding is of two folds. Firstly, it is used to restrain the model from predicting implausible pose patterns by constraining the solution space to follow a particular prior distribution. Secondly, the same embedding space is used as the shared latent representation to define higher order relations for the cross-modal alignment.

We train an AAE [11], $\{\mathtt{E}_\mathtt{p}, \mathtt{D}_\mathtt{p}\}$ on the unpaired 3D pose samples, $y \in \mathtt{Y}$. Its latent embedding $z \in \mathcal{Z}$ follows a uniform prior distribution $\mathbb{U}[-1, 1]^{32}$. The generative pose decoder $\mathtt{D}_\mathtt{p}$ maps any random vector $z \in \mathbb{U}[-1, 1]^{32}$ to a continuous human pose manifold (see Fig. 2A). Note that, a simple *tanh* non-linearity on the neural output of the *image-to-latent* model ensures decoding of plausible pose pattern (see Fig. 2C), thus constraining the solution space. Further, we minimize the modeling burden

Table 3: Assets and the corresponding Licenses

| Asset used | License | Links |
|---|---|---|
| CMU MoCap [1] | CC BY-ND 4.0 | Dataset, License |
| Human3.6M [7] | Custom, for research purposes | Dataset, License |
| MPI-INF-3DHP [13] | Custom, for research purposes | Dataset, License |
| Sports-1M [8] | CC BY 3.0 License | Dataset, License |
| SURREAL [17] | Custom, for research purposes | Dataset, License |
| 3DPW [18] | Custom, for research purposes | Dataset, License |

Table 4: Specifications of the machine used for training the complete pipeline.

| CPU | GPU | RAM | VRAM | CUDA |
|---|---|---|---|---|
| Intel Xeom E5-2689 | NVIDIA Tesla V100 | 256GB GB | 32 GB | V11.0 |

of the auto-encoder by decomposing the raw root-relative 3D locations into parent-relative local unit vectors (*i.e.* view-invariant). Here, the view-variant information is retained in the coordinates of the *neck*, *left-hip*, and *right-hip* joints with *pelvis* being their parent, as presented in the original root-relative system. Thus, the neural output of the decoder $D_p$ is required to be passed through a forward-kinematic [19] module (FK in Fig. 2A) to obtain the final root-relative 3D pose. Here, the pose embedding is trained to model both rigid view-variations and non-rigid 3D articulations.

**b) Motion embedding.** Next, we train the motion embedding, $v \in \mathcal{V}$ as the intermediate representation of a recurrent auto-encoder $\{E_m, D_m\}$. We prepare fixed-length pose sequences, $Y \in \widetilde{\mathcal{Y}}$ to be used as the motion samples (see Fig. 2B). Unlike auto-regressive models [20, 12], we do not feed chained input to motion decoder $D_m$. As discussed before, the purpose of the motion embedding is to lay a suitable ground to formalize higher order temporal relations among entities in the shared latent space, $\mathcal{Z}$. Thus, the recurrent encoder operates on the sequence of pose embeddings which is obtained by passing the pose sequences through the frozen pose encoder, *i.e.* $Z = E_p(Y)$. The output of the motion encoder, $v = E_m(Z)$ is fed as the only input to initialize the hidden states of the motion decoder $D_m$. $D_m$ outputs a sequence of reconstructed pose embeddings which are then fed to the pose decoder to obtain the reconstructed pose sequence.

**c) Pre-training on labeled source data.** From the perspective of Unsupervised Domain Adaptation [4, 16], the SURREAL dataset can be seen as a labeled, synthetic source domain. The source-specific *image-to-latent* model, $G^s$ is trained by minimizing $\|D_p \circ G^s(x^s) - y^s\|$, where $\circ$ denotes functional composition. Accordingly, our next objective is to adapt the source-trained model to make it work for real images $x \in \mathcal{X}$ which are sampled from the unpaired in-the-wild videos, *i.e.*, the unlabeled target domain. During adaptation we learn a target-specific [10] *image-to-latent* mapping $G$ which is initialized from $G^s$. However, only a minimal set of parameters of $G^s$ are allowed to learn target-specific mapping [14] whereas others are kept frozen from source initialization.

# 6 Qualitative analysis

We extend the qualitative results for 3D human pose estimation from the main paper and show extensive results on in-the-wild datasets in Fig. 5 and Fig. 4. The results shown are using *Ours(SH→W)* model as described in the main paper. The datasets used are highly diverse in foreground appearance, poses, and backgrounds. Even under such variations, the model generalizes to complex athletic poses establishing the effectiveness of the learned pose embedding. The last row, separated by the horizontal line shows the failure cases. Highly cluttered background, rarely seen poses, and multi-level inter-limb occlusion pose a challenge for the model. Despite this, the predicted poses still look plausible and quite close to the ground truth in most of the cases.

# 7 Reference to code and assets

We utilize the machine specifications mentioned in Table 4 for training and testing all our models. We implement the model using the open source TensorFlow 1.0 [2] framework in Python3.8. Table 3 lists

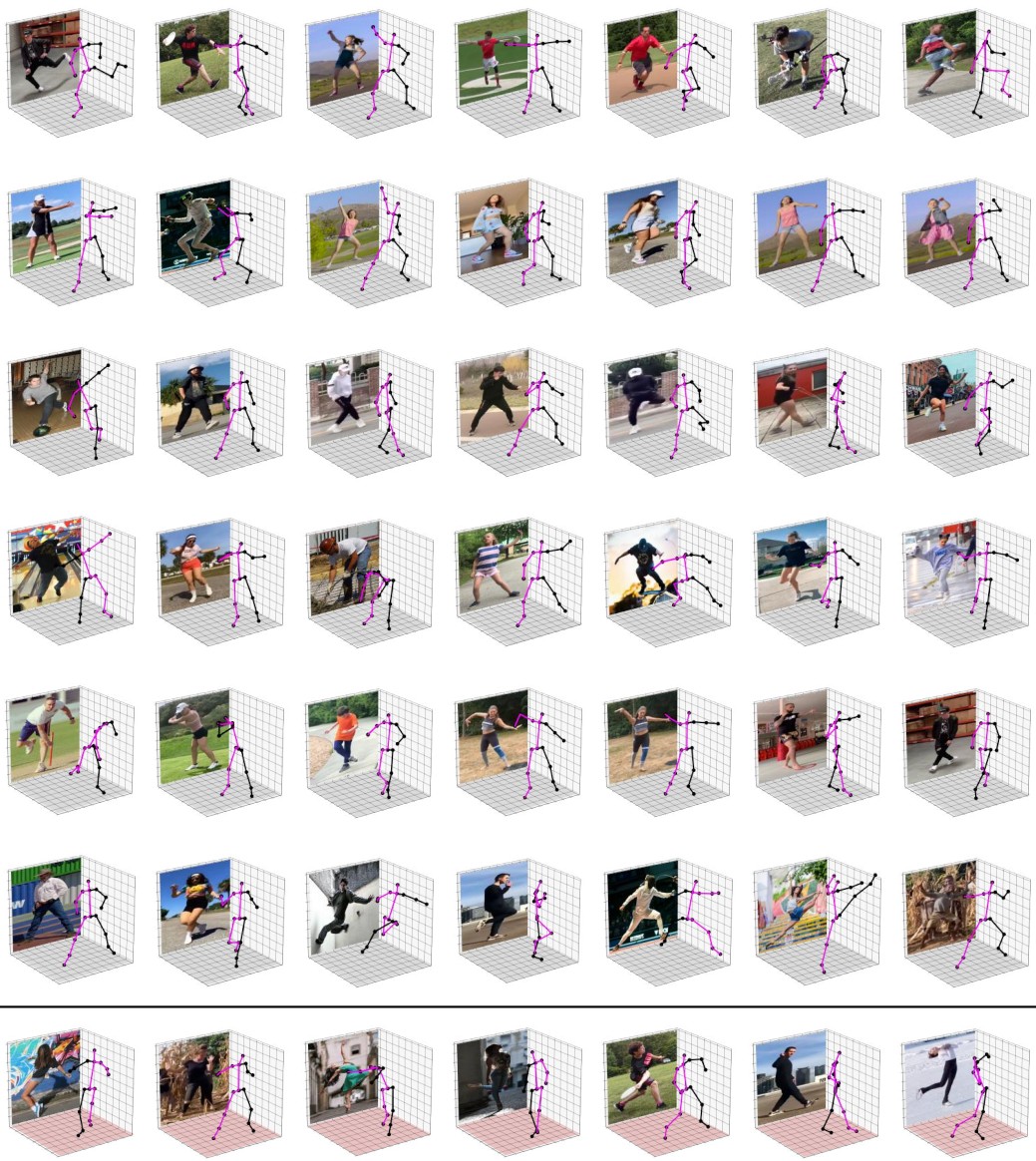

Figure 5: Qualitative analysis. The 3D pose estimation results are shown on Web-dataset, which consists of highly diverse pose, background, and foreground apparel. Predictions shown are obtained via *Ours(SH→W)*. The failure cases (rare poses, inter-limb occlusion, and high foreground-background clutter) are highlighted by results with red bases, below the horizontal bar.

licenses of the assets used in our research. A sample codebase of the proposed approach is provided at our project page[1].