# OpenReview forum: "Non-local Latent Relation Distillation for Self-Adaptive 3D Human Pose Estimation"
_NeurIPS.cc/2021/Conference — NeurIPS 2021 Poster_

### Official Review · Reviewer_jckK · 2021-07-15

**Rating:** 4
**Confidence:** 3

**Summary:**

The paper focuses on 3D human pose estimation in a scenario of domain adaptation. To avoid dataset-bias, the authors address the problem in a more restrict setting: any form of paired supervision or auxiliary cues should not be used.

The authors first use AAE modules to learn pose and motion latent spaces. Secondly, the authors train a pose encoder which tries to map the input image to the learned pose latent space. Finally, given the target-domain videos, the authors train the pose encoder together with the decoder of the pose AAE. The model is optimized by minimizing relational energies.

**Main Review:**

Strength
- This paper present a good solution for domain adaptive 3D human pose estimation. Based on the proposed setup, it is technically valid, and it should work well.
- Good figures to better illustrate the proposed idea and the learned embedding.
- Good ablation study of the use of different relational energies.
- The results are strong. In some cases, the proposed method even outperforms the multi-view approaches.


Weakness
- The presentation is very difficult to follow. This is probably due to the writing style of the authors. Most of the things are described in a very abstract manner. It is difficult for me to understand the proposed method in details. I spent hours to read the paper, but still I am not confident about all details in the proposed solution.  Also, I can't find strong connections with the cognitive science. The authors tend to introduce new terms, but I find them a bit distracted.

- The proposed method involves multi-stage training, but not end-to-end training. Each training stage includes a new model, and each would affect the final performance. Given many uncertainties, it is difficult to understand the performance. I would suggest to ablate each training stage, and show the effectiveness of each module.

- What is performance of the AAEs? What quality of the pose embedding is required? Same question for the motion embedding.

- Instead of training AAEs to generate pre-learned latent feature, the authors may want to use SMPL pose parameters as an alternative choice of the latent feature. Since SMPL parameters are also pre-trained and commonly used in the literature, this may serve as a good baseline to better promote the proposed method.

- Why should we use a pre-learned output pose embedding, but not fully end-to-end learned embedding? The pre-learned embedding may lead to sub-optimal performance. The design choice of the proposed method is not well motivated.

- Lack of baselines: One important baseline is to use pseudo pairs, and perform distillation, as the authors described in line181. Another baseline is domain adaptation with adversarial training. It is better to provide concrete performance comparisons to justify the proposed method.

- The authors argue that they do not use paired supervision or auxiliary cues to avoid dataset bias. However, the proposed method replies on Faster RCNN, which may imply it still require supervised pre-training on large number of paired training data. The authors may want to add more discussions about how they avoid the dataset bias.


Overall:
- The paper introduces an interesting method for human pose estimation in the setting of domain adaptation. However, due to bad writing, it is very difficult to follow (at least to me). Many things remain unclear in the paper. Since the target scenario is relatively different from prior works, I would suggest the authors to build their own baselines, in order to better justify the proposed solution. After careful consideration, I think this paper is not ready for a publication.


*******Post rebuttal*********
Thank you for the detailed response. My overall comments remain the same. I think the presentation is difficult to follow, and requires several revisions.


**Time Spent Reviewing:**

72

---

> ### Author Response · Authors · 2021-08-10
> **Response to Reviewer jckK**
>
> We thank the reviewer for the constructive feedback. We appreciate that the reviewer finds that our work presents a good solution supported by good figures, good ablation, and strong results. We address the reviewer’s concerns below.
>
> 1. ***Presentation is difficult to follow -*** Our quest to develop an effective solution for the challenging problem setting led us to draw motivation from a wide range of literature beyond pose estimation, such as Domain Adaptation and Knowledge Distillation. Given the restricted page limit, we have tried our best to present different perspectives under well-structured logical flow via subsections (Sec 3) and paragraph headings (Sec 2) while moving the details of the implementation to Suppl. We request the reviewer to refer Suppl-Sec 4 and Suppl-Fig 2 for a better understanding. We assure to make the presentation more lucid and reader-friendly. Further, we will include the [attached figure](https://drive.google.com/file/d/1K3HY2m0kzt-t6CtL4tYnuaMSQisnlBqG/view?usp=sharing) (refer response #2 of Reviewer-UoE5) in the Introduction, illustrating our key novelty of using non-local relations.
>
> * [18, 20] advocate that a key factor in self-supervised human learning is the acquisition of new knowledge by relating
> entities. The proposed non-location relation distillation draws motivation from the same idea, i.e., by relating non-local entities via relations such as “flip” (in pose space) and “backward” (in motion space). We will discuss it more explicitly in the revised draft.
>
> 2. ***Ablation over each module -*** We would like to clarify that the proposed self-adaptive losses are defined on top of the pre-learned modules. Thus, *the ablation study provided in Table 5 already addresses the reviewer’s concern* regarding the ablation of each training stage and the corresponding module involvements. We elaborate it in the following table.
>
> | Ablation                                  |   Modules involved                               | MPJPE |
> |  -                                             |  :-:                                                         |        -     |
> |      Source-only                       |      $\mathtt{G}$, $\mathtt{D_p}$        |  209.6  |
> |      +$\mathcal{L}_{LCR}$        |      $\mathtt{G}$, $\mathtt{D_p}$        |   193.4  |
> |      +$\mathcal{L}_{HCR}$        |      + $\mathtt{E_m}$                           | 172.1 |
> |      +$\mathcal{L}_1^z$           |      + $\mathtt{T}^z_1$                         | 141.2 |
> |      +$\mathcal{L}_1^v$            |      + $\mathtt{T}^v_1$                         | 95.6 |
> |      +$\mathcal{L}_2^v$            |      +  $\mathtt{T}^v_2$                        | 86.8 |
>
> 3. ***Performance of AAEs -*** Both the pose and motion AAE are generative models, thus there is no straightforward way to comprehend their performance towards the goal task. The reviewer may refer to the architectures of the pose and motion AAEs in Suppl-Sec. 4 to get an idea about the depth (or capacity) of the AAE architecture in order to relate it with the quality.
>
> 4. ***SMPL pose parameters as an alternative choice -*** We would like to clarify that the use of SMPL parameters as latent does not restrict predictions to plausible poses. Prior-arts using SMPL [29, 30] require additional adversarial discriminator loss to impose the pose prior, which greatly destabilizes the training process (particularly in the absence of cross-modal pairs), resulting in convergence to degenerate solutions. *Please refer Suppl L88-92 which explains how a simple tanh nonlinearity on the output of $\mathtt{G}$ restricts the pose output within the realm of natural pose distribution*. Thus, SMPL can not be used as a replacement for the pre-learned AAE. Nonetheless, using SMPL based latent does not provide any new insight regarding the primary focus of this work, i.e., developing relation distillation-based self-adaptive objectives.
>
> 5. ***Why use a pre-learned output pose embedding? -*** We would like to clarify that, we freeze $\mathtt{D_p}$ while training $\mathtt{G}$, as it is crucial in order to regularize the unsupervised adaptation thereby avoiding convergence to degenerate solutions in the absence of cross-modal pairs.
>
> * Updating $\mathtt{D_p}$ while training $\mathtt{G}$ requires us to impose an additional adversarial discriminator loss (as used in HMR [29]) in order to uphold its ability to decode plausible pose predictions. Note that, updating $\mathtt{D_p}$ would update the manifold structure of the embedding space. Further, as the relation transformers  ($\mathtt{T}^z_1$, $\mathtt{T}^v_1$, $\mathtt{T}^v_2$) operate on the frozen latent embeddings, the pre-learned relations transformer networks (used to define the relation distillation objective) would no longer be useful and are required to be updated alongside. Such unconstrained optimization greatly destabilizes the self-adaptive process (in the absence of cross-modal pairs) and worsens the performance to a great extent (refer the table in response #2 of Reviewer-fswa). Note that the proposed self-adaptation step does not involve any complex adversarial loss component, which greatly simplifies the training procedure.
>
>
> 6. ***Lack of baselines -*** Thank you for the suggestion. The following table reports a comparison of self-adaptation performance using baseline adaptation techniques (used for object recognition tasks) against the proposed procedure. Sec 3.2 already discusses the reason behind the significant performance gap. We will include it in the revised draft.
>
> |               |            Baseline Method        | MPJPE |
> |      -       |                     :-:                        |       -      |
> |      B1       |         Pre-adaptation                       |       209.6     |
> |      B2       |         DA via adv. Learning [73]       |       206.8     |
> |      B3      |          Using Pseudo-pairs                |       201.3     |
> |                |          Ours(S→H)                           | 86.8   |
>
> 7. ***Reliance on Faster RCNN -*** We understand the reviewer’s concern. We use Faster R-CNN to segregate human vs non-human video frames in order to obtain a pruned Web-dataset and this should be considered as a preprocessing step. Almost all the self-supervised or unsupervised works in literature, focusing on a) object recognition [53, 12] (even in ImageNet the objects are well centered) or b) human pose estimation [36, 16, 11, 60, 86], use such detectors to obtain object-centered image frames. This is well accepted by the community as a standard pre-processing step. To support this, we quote from [60]: “It is common practice for human pose estimation algorithms to first crop the subject of interest to factor out scale and global position ... We also do this and use the crop information provided in the training datasets.” We will discuss it in the revised draft.

---

### Official Review · Reviewer_vt6r · 2021-07-16

**Rating:** 6
**Confidence:** 5

**Summary:**

This work regards monocular 3D human pose estimation as a domain adaptation problem to solve the problem of insufficient outdoor training data. The framework is composed of an image-to-latent encoder, which is pre-trained on the source domain and adapted to the target domain using unlabeled videos, and a latent-to-pose decoder, which is trained on unpaired 3D poses. During the encoder adaptation, non-local relations are utilized to improve the performance, which is the main contribution.


**Ethical Concerns:**

No ethical issues found.

**Limitations And Societal Impact:**

Yes. Both limitation and societal impact is addressed in this paper.

**Main Review:**

Pros:
Non-local relations are involved in the domain adaptation, which is novel and proved to be effective by experiments.
Achieve SOTA results on several datasets under the semi-supervised and unsupervised settings.
Well organized and easy to follow.

Cons:
There is an important hypothesis in this paper: Large embedding distance of non-local motion relations is beneficial to the domain adaptation. However, this hypothesis is not carefully verified. An ablation study with different embedding distances is needed, which can be achieved with in-plane rotation with different angles.

When compared with other methods, the training data is different, which makes the comparison unfair.

In the experiment, larger rotation angle results in a larger embedding distance, so is it possible to achieve a larger embedding distance than flipping by increasing rotation angle or combining flipping and rotation to a new motion? Will this further improve the performance?

Some Typos:
In table5, the metric should be PA-MPJPE rather than MPJPE. And we recommend showing both MPJPE and PA-MPJPE.


**Time Spent Reviewing:**

5 hours.

---

> ### Author Response · Authors · 2021-08-10
> **Response to Reviewer vt6r**
>
> We thank the reviewer for the comments and questions. We are glad that the reviewer finds our work novel, well-organized, and easy to follow and acknowledges the use of “non-local relations” as our key contribution. We address the reviewer’s concerns below.
>
> 1. ***Ablation study to verify the hypothesis -*** Thank you for the suggestion. An ablation with increasing in-plane rotation angles would be a strong support to our key hypothesis. The following table reports MPJPE (lower is better) with *InPlane-*$\theta$ (2nd row) and $\theta$ -*backwards* (3rd row) as the lower-order (pose space) and higher-order (motion space) relations respectively (in the settings of Table 5). In both cases, $\theta$ is varied as 10, 25, 50, 75, 100, 125. In the last column, we compare the results with “flip” and “flip-backwards” in the 2nd and 3rd rows respectively.
>
> | InPlane-$\theta$ $\rightarrow$ | $10^\circ$ | $25^\circ$ | $50^\circ$ | $75^\circ$ | $100^\circ$ | $125^\circ$ | Flip | Flip+InPlane-$50^\circ$ |
> | - | - | - | - | - | - | - | - | - |
> | Lower-order | 179.1 | 170.3 | 154.4 | 163.9 | 159.2 | 158.4 | 141.2 | **139.7** |
> | Higher-order (+backwards) | 130.5 | 123.2 | 110.1 | 115.6 | 114.1 | 115.4 | 95.6 | **91.8** |
>
> * ***Observation -*** We see an increase in adaptation performance with an increase in in-plane rotation for $\theta=10, 25, 50$. However, with a further increase in $\theta$ (i.e., $75, 100, 125$) the performance seems to be saturating at a degraded level. The prime reason behind this behavior is attributed to the fact that pose samples with $\theta>50$ are quite rare (fall in the low probability region of the latent pose space). For example, images depicting poses with the spine parallel to the ground or headstand are very rare. However, the probability of encountering images with a flipped pose is quite high, and both the original and flipped poses fall in the high probability region of the latent pose space.
>
> * ***With Flip+InPlane -*** We thank the reviewer for the valuable insight. A non-local relation with a combination of Flip and a suitable In-Plane rotation, i.e.,  *Flip+InPlane-*$50^\circ$ yields the best performance (in both lower and higher-order cases) beyond using just Flip or In-Plane rotation. We will include it in the revision.
>
> * ***In summary***, the hypothesis is valid (and more effective) as long as both the relational association candidates fall in the high-probability regions of the latent pose space. We will discuss it in more detail in the revision.
>
> 2. ***Unfair comparison due to differences in training data -*** We understand the reviewer’s concern. However, training such models from scratch takes a considerable amount of time (several days). Comparisons with differences in training data are generally accepted in the community as long as the advantages and disadvantages are clearly reported. For instance, Table 1 in Doersch et al. [16] compares [16] (uses custom in-house data) against HMR [29] and Martinez et al. [44] despite having significant training data differences. Similarly, even though our training data is different, we are at a disadvantage at several places. For example:
>
> * **a)** In Table 2, [27] accesses multi-view supervision and the labeled MPII dataset for semi-sup results which Ours (S→H,Semi) does not use. **b)** In Table 2, Kundu et al. [36] use an additional YouTube dataset. In Table 3, [87] uses labeled MPII data unlike in Ours(H→M). **c)** In Table 4, unlike Ours(SH→W), [30] additionally uses PennAction and NBA whereas [29] additionally uses LSP, MPII, and MS-COCO. This clearly highlights the significance of our performance, as we do not rely on any in-the-wild, labeled 2D pose data.
>
> 3. ***Typos -*** Thank you for pointing out the typo in Table 5 . We will rectify it and include both MPJPE and PA-MPJPE metrics in the revised draft.

---

> > ### Comment · Reviewer_vt6r · 2021-08-23
> > **Feedback to the rebuttal**
> >
> > Thanks for your reply.
> >
> > In the reply, my concerns are well solved. The new ablative experiment verified the key hypothesis, and the advantages of the method are also reported even with difference in the training data. Taking the novelty and effectiveness of the method into consideration, I think this is a good paper which provides a new idea for 3D human pose estimation.
> >
> > Above all, I'd like to raise the store to '6: Marginally above the acceptance threshold'.

---

> > > ### Author Response · Authors · 2021-08-27
> > > **Thanks for the feedback**
> > >
> > > Thank you for the encouraging feedback. We request you to update the score in the original review.

---

### Official Review · Reviewer_UoE5 · 2021-07-19

**Rating:** 6
**Confidence:** 4

**Summary:**

This paper proposed to learn to predict the 3D human pose for real-world images via adapting the model from the source data (label constraints) to unlabeled data. The data used for training contains: (1) the labeled source domain data (e.g. human3.6m or surreal); (2) real human video sequence without pose annotations; and (3) motion capture data (3D pose without images). (2) and (3) are unpaired. The authors proposed to devise: (a) image-to-latent; (b) pose-to-latent; and (c) latent-to-pose, to learn and predict 3D poses. In particular, (a) and (c) forms the test time predictions for the real-world target domain, while (b) and (c) constitutes an auto-encoder. The core problems in here is that the latent space in (ac) and (bc) might be mismatched. The authors proposed a smart way to make sure they are aligned (see Strength below) thanks to the contrastive loss (for setting up latent space non-local grouping rules) as well as the loss from the source domain data for regularization to make sure the two latent domains align with each other.

**Limitations And Societal Impact:**

The authors have addressed them well in the paper.

**Main Review:**

Strength:

I really appreciate the main novelty of the paper. To put the spark of the paper in one sentence: the paper proposed an innovated self supervised method to align the human pose latent space from unpaired image-to-latent and the pose-to-latent, via enforcing several latent space rules that is available for unpaired image-to-latent and pose-to-latent domains (e.g. symmetric poses should be close, named as the "positive non-local coupling" by the author). The loss from the source data (D^s) would serve as the extra constraint to make sure the two domains align. As long as this is the first time to propose this type of the human-pose-specific rules of latent space (I have not checked existing literature thoroughly), I would be quite positive on this paper. The experimental part provides fairly good results (while a notch below very strong visual results, see below weakness part).

Weaknesses:

1. I have moderate concerns to the experimental part, as the visual results and comparisons are not particularly exciting. Comparisons to several SoTA seems missing (e.g. Frank [a] and SMPL-X [b]) although these are not self-supervised or domain adaptation approaches. I wish the authors can also provide prediction overlay and their rotated 3D visual results. The current given visualization seems quite difficult to judge the accuracy. Besides, since one of the major domain is the video and pose sequence, I hope the authors can provide results for several human video clips.

2. The writing of the paper seems not smooth to me, and I need to take efforts to reach the spark of the paper. It seems the introduction part did not well express the core novelty of the proposed framework. I would suggest the authors add one small illustrative figures in the intro part, particularly illustrating the non-local positive coupling is available in both the image-to-latent and the pose-to-latent mappings. I hope that can help the readers to get the message very quickly.

3. Small issue: It seems the term "non-local" used in this paper is irrelevant to that in [c]. I wish the authors can either provide a clarification in the related works section, or use a slightly different name.

[a] Joo, Hanbyul, Tomas Simon, and Yaser Sheikh. "Total capture: A 3d deformation model for tracking faces, hands, and bodies." In Proceedings of the IEEE conference on computer vision and pattern recognition, pp. 8320-8329. 2018.

[b] Pavlakos, Georgios, Vasileios Choutas, Nima Ghorbani, Timo Bolkart, Ahmed AA Osman, Dimitrios Tzionas, and Michael J. Black. "Expressive body capture: 3d hands, face, and body from a single image." In Proceedings of the IEEE/CVF Conference on Computer Vision and Pattern Recognition, pp. 10975-10985. 2019.

[c] Wang, Xiaolong, Ross Girshick, Abhinav Gupta, and Kaiming He. "Non-local neural networks." In Proceedings of the IEEE conference on computer vision and pattern recognition, pp. 7794-7803. 2018.

Considering the good novelty and the fair experimental results of the paper, I rate this work as "borderline accept".

** Post Rebuttal ** After reading all the reviews and the authors' responses, I believe this paper provided innovated insights, while all the reviewers mentioned the clarity concern of the paper. I would keep my original rating considering all these factors.


**Time Spent Reviewing:**

6

---

> ### Author Response · Authors · 2021-08-10
> **Response to Reviewer UoE5**
>
> We thank the reviewer for the constructive comments. We appreciate that the reviewer identifies our work to be an innovative self-supervised method and acknowledges our key novelty of using “positive non-local coupling” for self-adaptation.
>
> We would like to mention that, to the best of our knowledge, we are the first to propose such human-pose-specific rules of latent space for self-adaptation. Both the problem setting and solution are not being explored in any prior arts.
>
> Below we address the reviewer’s concerns.
>
> 1. ***Comparisons to SoTA missing -*** Thank you for pointing out the papers [a,b]. We will discuss these in the related works. Both [a] and [b] build on the parametric SMPL body model and extend it to express finer movements such as hand gestures and facial expressions, which is different from the primary focus of our paper. We would like to mention that [a,b] do not fit into our current quantitative comparison as the evaluation criterion, i.e. focus dataset and metric do not align.
>
> 2. ***Improving visual comparisons -*** Thank you for the suggestions. We understand that better visualization is required to judge the accuracy on datasets without 3D pose GT (i.e., without quantitative comparison), such as Web-dataset and LSP. The improved visualizations are presented at this [Fig-link](https://drive.google.com/file/d/1VUGtnz_kEapRsrPZogCzCCcCTUzRSxYS/view?usp=sharing). We use the same settings as in Fig.4 (main paper).
>
> 3. ***Illustrative figure for Introduction -*** Thank you for the suggestion to improve the Introduction. Please see the [attached link](https://drive.google.com/file/d/1K3HY2m0kzt-t6CtL4tYnuaMSQisnlBqG/view?usp=sharing) containing an illustrative figure that better explains our key novelty. We will include this in the Introduction and refer to its different parts in Sec 1 for a smoother understanding of the logical flow and to highlight our key novelty.
>
> 4. ***Conflicting use of the term “non-local” -*** Thank you for pointing out the conflicting use of the term “non-local” in [c]. We understand that this may cause confusion. We will clarify this in the “Related Works” section.

---

### Official Review · Reviewer_fswa · 2021-07-20

**Rating:** 6
**Confidence:** 2

**Summary:**

## Summary

Paper proposes a self-adaptation framework to align the unpaired samples of 3D pose and video frames to solve 3D human pose estimation from in-the-wild images. This cross-modal alignment is realized through a relation distillation scheme. The proposed approach achieves compelling results on 3D human pose datasets among self-supervised and unsupervised methods.

**Limitations And Societal Impact:**

Yes, the limitations and societal impact are discussed in the paper.

**Main Review:**

## Strengths

- Paper tackles a real problem of scarcity of 3D pose labels for in-the-wild unconstrained images. They propose a self-adaptation method that helps to utilize web videos and large-scale motion capture datasets both of which are available in abundance.

- Proposed domain adaptation approach sounds novel. Motivation and implementation make sense to be able to perform unsupervised adaptation.

- Experiments are detailed. Three major human pose estimation datasets are used for the model's performance. Ablation experiments show that propose the contribution of each relation distillation term into the final performance.

## Weaknesses

- Paper writing is quite hard to follow. Especially in the intro, there are lots of technical terms which is not familiar to the human pose estimation community. These are mostly domain adaptation or knowledge distillation terms and they should be introduced in a logical flow to the reader.

- CMU mocap dataset is chosen as the unpaired 3D pose dataset. AMASS dataset is much larger in terms of quantity, the number of subjects, pose variation, and actions performed. It also includes CMU as a subset. So, AMASS could be a better choice for the proposed method.

- A motivation of the proposed approach is overcoming the dataset bias. However, the Dp is kept frozen when used with G. This means that the manifold learned/encoded by Z is biased towards the unpaired 3D pose dataset which is CMU mocap in this case. I think this is a major limitation of the current method and this should be tackled to overcome the dataset bias.

### Post Rebuttal

Authors addressed my comments in a satisfactory way. Considering the other reviews and author's responses to them I would like to keep my initial score of "6: Marginally above the acceptance threshold".

**Time Spent Reviewing:**

6 hours

---

> ### Author Response · Authors · 2021-08-10
> **Response to Reviewer fswa**
>
> We thank the reviewer for the valuable feedback. We are encouraged that the reviewer believes we tackle a real problem with a novel self-adaptive approach supported by detailed experiments. Below we address the reviewer’s concerns.
>
> 1. ***Writing is hard to follow -*** Our quest to develop an effective solution for this challenging problem setting led us to draw motivation from a wide range of literature beyond pose estimation, such as Domain Adaptation and Knowledge Distillation. Although the technical terms are well elaborated in Sec 2 and in Sec 3.2, we missed to formally introduce these in Sec 1. We assure to update it in order to make it more lucid and reader-friendly. Further, we will include the [attached figure](https://drive.google.com/file/d/1K3HY2m0kzt-t6CtL4tYnuaMSQisnlBqG/view?usp=sharing) (refer response #3 of Reviewer-UoE5) in the Introduction, illustrating our key novelty of using non-local relations.
>
> 2. ***AMASS would be a better choice -*** Thank you for the suggestion. Using AMASS in place of CMU-MoCap would definitely improve our performance as depicted in the table below. We have used CMU-MoCap for a fair comparison as it has already been used in several prior arts [29, 36]. We will include this in our revised draft.
>
> | Methods | Unpaired 3D dataset | MPJPE ($\downarrow$) on Human3.6M |
> |       -       |                  -                |                   -                    |
> | Kundu et al. [36] | CMU-MoCap | 99.2 |
> | Ours(S→H) | + while updating $\mathtt{D_p}$| Does not converge |
> | Ours(S→H) | CMU-MoCap | 86.8 |
> | Ours(S→H) | AMASS | 84.1 |
> |  |  |  **PA-MPJPE ($\downarrow$) on 3DHP**
> | Kundu et al. [36] | CMU MoCap | 103.8 |
> | Ours(SH→W) | CMU-MoCap | 95.2 |
> | Ours(SH→W) | AMASS | 91.0 |
>
> 3. ***Freezing $\mathtt{D_p}$ biases the $Z$ manifold -*** We freeze $\mathtt{D_p}$ while training $\mathtt{G}$ as it is crucial to regularize the unsupervised adaptation to avoid degenerate solutions (or mode-collapse).
>
> * Updating $\mathtt{D_p}$ while training $\mathtt{G}$ requires us to impose an additional adversarial discriminator loss (as used in HMR [29]) in order to uphold its ability to decode plausible pose predictions. Note that, updating $\mathtt{D_p}$ would update the manifold structure of the embedding space. Further, as the relation transformers ($\mathtt{T}^z_1$, $\mathtt{T}^v_1$, $\mathtt{T}^v_2$) operate on the frozen latent embeddings, the pre-learned relations transformer networks (used to define the relation distillation objective) would no longer be useful and are required to be updated alongside. Such unconstrained optimization greatly destabilizes the self-adaptive process (in the absence of cross-modal pairs) and degrades the performance significantly. Note that the proposed self-adaptation step does not involve any complex adversarial loss component, which greatly simplifies the training procedure.
>
> 4. ***Overcoming output-space dataset bias -*** This can only be addressed in the presence of cross-modal pairs which is outside the focus of this work. The proposed approach caters well to input-space dataset bias. We thank the reviewer for pointing this out. Further, as depicted in the table above, incorporating a larger and more diverse unpaired 3D pose dataset, such as AMASS, could be a viable alternative to address output-space dataset bias.

---

### Decision · Program_Chairs · 2021-09-27

**Decision:**

Accept (Poster)

**Comment:**

This paper studies the problem of 3D human pose estimation by combining information from labeled sources and unlabeled data in the wild. The paper received mixed reviews, in the beginning, tending near borderline. The reviewers' major concern was regarding the presentation clarity. Multiple reviewers found it hard to follow. The authors provided a rebuttal that addressed some of the reviewers' concerns and promise to improve the presentation. The paper was discussed and most reviewers responded to the rebuttal. One of the reviewers raised their rating but one reviewer did not. The main complaint is still around the writing and presentation. AC agrees with the reviewers that the paper has a good contribution to be accepted and urges the authors to look at the reviewers' feedback, incorporate their comments and clarify the writing as promised in the camera-ready.